# MIND YOUR ENTROPY: FROM MAXIMUM ENTROPY TO TRAJECTORY ENTROPY-CONSTRAINED RL

## ABSTRACT

Maximum entropy has become a mainstream off-policy reinforcement learning (RL) framework for balancing exploitation and exploration. However, two bottlenecks still limit further performance improvement: *(1) non-stationary Q-value estimation* caused by jointly injecting entropy and updating its weighting parameter, i.e., temperature; and *(2) short-sighted local entropy tuning* that adjusts temperature only according to the current single-step entropy, without considering the effect of cumulative entropy over time. In this paper, we extend maximum entropy framework by proposing a trajectory entropy-constrained reinforcement learning (TECRL) framework to address these two challenges. Within this framework, we first separately learn two Q-functions, one associated with reward and the other with entropy, ensuring clean and stable value targets unaffected by temperature updates. Then, the dedicated entropy Q-function, explicitly quantifying the expected cumulative entropy, enables us to enforce a trajectory entropy constraint and consequently control the policy's long-term stochasticity. Building on this TECRL framework, we develop a practical off-policy algorithm, DSAC-E, by extending the state-of-the-art distributional soft actor-critic with three refinements (DSAC-T). Empirical results on the OpenAI Gym benchmark demonstrate that our DSAC-E can achieve higher returns and better stability.

## 1 INTRODUCTION

Balancing exploration and exploitation remains a central challenge in reinforcement learning (RL) (Sutton & Barto, 2018; Li, 2023). To address this, off-policy methods have leveraged the maximum entropy principle, which encourages agents to act as randomly as possible while still optimizing for high returns (Wang et al., 2022; Haarnoja et al., 2017). By augmenting the objective with a temperature-weighted entropy term, algorithms like Soft Actor-Critic (SAC) (Haarnoja et al., 2018a) and its distributional variant DSAC (Duan et al., 2021; 2025) have achieved state-of-the-art performance on continuous control benchmarks like MuJoCo, proving to be highly effective and robust (Eysenbach & Levine, 2022).

However, a fixed temperature parameter can lead to a policy that is either excessively stochastic or unnecessarily deterministic (Rawlik et al., 2012). This is because a single temperature value cannot optimally balance exploration and exploitation across all phases of training; a high temperature may hinder convergence, while a low temperature can lead to premature exploitation of a suboptimal solution (Fox et al., 2016). To mitigate this issue, modern maximum entropy RL incorporates an automated temperature adjustment mechanism (Haarnoja et al., 2018b). Using the policy's current per-step entropy as a feedback signal, this mechanism dynamically tunes the temperature throughout training, aligning it with a predefined target. Therefore, it ensures that a desired level of stochasticity is maintained across all situations (Hazan et al., 2019).

Despite the remarkable empirical success, maximum entropy methods still face two critical bottlenecks that hinder further progress. (1) The first issue is *non-stationary Q-value estimation*, which stems from the tight coupling of reward and entropy (Schulman et al., 2017a). Since the temperature parameter is updated simultaneously, the injected temperature-weighted entropy term is directly altering the Q-value targets, causing them to become non-stationary. This process can destabilize value learning and ultimately undermine policy optimization (Lillicrap et al., 2016). We acknowledge that the bootstrapping update mechanism of Q-values contributes significantly to the non-stationarity

in RL. In this context, we highlight that the coupling of reward and entropy is another crucial contributing factor, and our method can effectively address and eliminate this factor. (2) Second, and perhaps more fundamentally, while some works have explored constraining entropy (increase the temperature if the entropy at the current step falls below a target value, and decrease it otherwise), they all suffer from *short-sighted local entropy tuning* (Haarnoja et al., 2018b; Duan et al., 2021; 2025). By regulating only the local current-step entropy, these methods neglect the long-term influence of stochasticity over entire trajectories. More critically, why we say they are short-sighted is that they enforce a uniform entropy target for each current state, as if every situation demands the same degree of randomness. This one-size-fits-all assumption is overly restrictive and fails to account for the inherent variability in the dynamics of different states. Consequently, the actor update process is compromised, as it neglects the fact that effective exploration should take into consideration both the underlying system dynamics and the agent's learning progress (Tokic, 2010; Sun et al., 2022). This fundamental disconnect ignores the varying exploration needs of different situations.

The observed two bottlenecks naturally raise a question: *can we move beyond maximum entropy by directly and cleanly controlling what really matters—the cumulative entropy of the policy?* We argue the answer is yes by introducing a trajectory entropy-constrained (TEC) RL framework. To ensure a stable and interpretable learning process, our core innovation is to completely decouple the reward and entropy signals by learning two separate Q-functions. This separation ensures clean Q-value targets, and the dedicated entropy critic enables us to enforce a trajectory-level constraint on the policy's cumulative entropy. This design inherently breaks from traditional single-step restriction, enabling a more principled and long-term control of policy stochasticity.

To demonstrate the practical advantages of our framework, we introduce DSAC-E, an extension of the state-of-the-art Distributional Soft Actor-Critic with Three refinements (DSAC-T) algorithm (Duan et al., 2025). DSAC-E integrates the strengths of DSAC-T's distributional value estimation with our proposed trajectory entropy constraint. By decoupling the reward and entropy Q-values and adjusting the trajectory-level entropy budget, our DSAC-E achieves cleaner and more effective exploitation alongside more controllable exploration. Empirical results on the OpenAI Gym continuous control benchmark (Brockman et al., 2016) demonstrate that DSAC-E not only achieves superior final returns but also exhibits better training stability than strong maximum entropy baselines.

Our contributions are summarized in threefold:

- We identify and analyze the impact of two bottlenecks in conventional maximum entropy RL: *(1) non-stationary Q-value estimation* and *(2) short-sighted local entropy tuning*. These issues motivate us to execute reward-entropy separation (RES) and trajectory-level entropy constraint (TEC);

- To address these two identified bottlenecks, we propose the TECRL framework. Within this framework, we first eliminate the *(1) non-stationary Q-value estimation* problem by decoupling reward and entropy signals into two separate critics, while temperature is excluded from the learning processes of both critics. Then the dedicated entropy critic allows us to enforce a trajectory-level entropy constraint, thereby overcoming the issue of *(2) short-sighted local entropy tuning*. Furthermore, we provide a rigorous theoretical analysis demonstrating that appropriately selecting a trajectory entropy budget can yield a higher performance bound;

- We introduce DSAC-E, a practical instantiation of our TECRL framework built on DSAC-T, the state-of-the-art maximum entropy algorithm. Through this instantiation, we demonstrate that our framework enables superior performance on complex continuous control tasks.

## 2 PRELIMINARIES

**Maximum entropy RL.** While standard RL seeks a policy that maximizes the expected accumulated return, maximum entropy RL (Haarnoja et al., 2017) extends this by adopting an objective function that incorporates a policy entropy term as

$$J_\pi = \mathop{\mathbb{E}}_{s_t \sim \rho_\pi} \Big[ \sum_{t=0}^{\infty} \gamma^t [r_t + \alpha \mathcal{H}(\pi(\cdot|s_t))] \Big], \tag{1}$$

where $\gamma \in (0, 1)$ is the discount factor, $\rho_t$ is the state visitation distribution, $\alpha$ is the temperature coefficient, and the single-step policy entropy $\mathcal{H}$ is expressed as

$$\mathcal{H}(\pi(\cdot|s_t)) = \mathop{\mathbb{E}}_{a_t \sim \pi(\cdot|s_t)} \big[ - \log \pi(a_t|s_t) \big]. \tag{2}$$

The optimal policy can be derived through a maximum entropy variant of policy iteration, commonly known as soft policy iteration (Wang et al., 2022). This iterative process alternates between two key stages: (1) soft policy evaluation (PEV) and (2) soft policy improvement (PIM).

In soft PEV, provided a policy $\pi$, for a given policy $\pi$, we can apply the soft Bellman operator $\mathcal{B}^\pi$ to learn the soft Q-value, as shown by the soft Bellman expectation equation:

$$\mathcal{B}^{\text{soft}}[Q^{\text{soft}}(s,a)] = r + \gamma \mathbb{E}_{s' \sim p, a' \sim \pi}[Q^{\text{soft}}(s',a') - \alpha \log \pi(a'|s')], \tag{3}$$

where the definition of soft Q-value is

$$Q^{\text{soft}}(s,a) = \mathbb{E}_\pi \left[ \sum_{t=0}^\infty \gamma^t r_t + \sum_{t=1}^\infty \gamma^t \alpha \mathcal{H}(\pi(\cdot|s_t)) \, \middle| \, s_0 = s, a_0 = a \right]. \tag{4}$$

One might ask why we write the reward and entropy signals as two separate summation terms. The reason is to highlight the difference in their starting indices. The reward signal is accumulated from the current time step, with a summation index of $t = 0$, while the policy entropy is accumulated from the next time step, with a summation index of $t = 1$. This difference is evident from the soft Bellman expectation equation in Eq. (3): the first term on the right-hand side, $r$, does not have a corresponding policy entropy term at the same time step. In fact, the missing current entropy $\mathcal{H}(\pi(\cdot|s_0))$ occurs in the subsequent soft PIM step.

In soft policy improvement (PIM), the goal is to find a new policy that outperforms the current policy. This is achieved by directly maximizing an entropy-augmented objective, a process equivalent to:

$$\pi_{\text{new}} = \arg\max_\pi \mathop{\mathbb{E}}_{s \sim \rho_\pi, a \sim \pi} \big[ Q^{\text{soft}}(s,a) - \alpha \log \pi(a|s) \big]. \tag{5}$$

The convergence of soft policy iteration to the optimal maximum entropy policy is a well-established result in the field, as shown by (Haarnoja et al., 2017).

**Temperature tuning.** A key advancement in the latest maximum entropy frameworks is the automatic management of the temperature parameter $\alpha$. Instead of being a fixed hyperparameter, $\alpha$ is treated as a learnable variable. The objective is to minimize

$$J(\alpha) = \mathbb{E}_{a_t \sim \pi} \big[ - \alpha \big( \log \pi(a_t|s_t) + \mathcal{H}_0 \big) \big], \tag{6}$$

where the default value of $\mathcal{H}_0$ is commonly set as $-\dim(\mathcal{A})$, i.e., the minus of the number of action dimensions. This mechanism achieves a dynamic balance between exploration and exploitation by maintaining the policy's local entropy close to a predefined target entropy $\mathcal{H}_0$ across all situations (Haarnoja et al., 2018a).

## 3 METHOD

### 3.1 TWO BOTTLENECKS OF MAXIMUM ENTROPY RL

Previously, we briefly introduced two bottlenecks that exist in the current maximum entropy RL framework. Now, combining with specific formulas, we will more formally and mathematically explain their origins and their impact on policy learning.

**(1) Non-stationary Q-value estimation.** In each soft PEV step, as shown in Eq. (3), the target value is calculated by

$$y_{\text{target}} = r(s,a) + \gamma[Q^{\text{soft}}(s',a') + \alpha \mathcal{H}(\pi(\cdot|s'))]. \tag{7}$$

When the temperature $\alpha$ is updated at the same time, the target value distribution shifts dynamically. This entanglement injects additional variance and bias into Q-value estimation, degrading subsequent policy improvement steps that rely on stable value predictions.

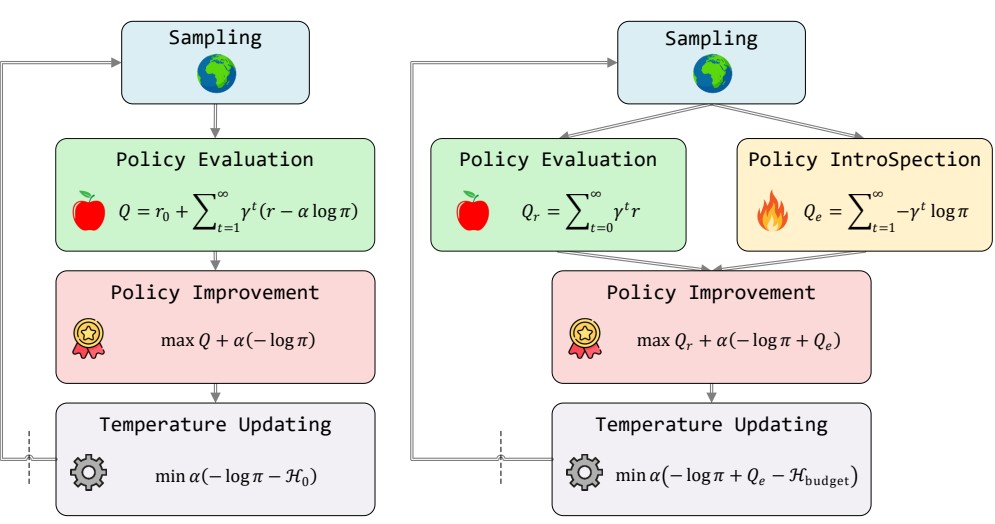

Figure 1: **Comparison** between standard maximum entropy RL (**left**) and our trajectory entropy-constrained (TEC) RL (**right**). Our TECRL framework comprises four key components: a reward-centric policy evaluation (PEV), an entropy-centric policy introspection (PIS), a policy improvement (PIM) that retains the exact soft policy objective, and a temperature update (TUP) tuning the temperature guided by the trajectory entropy constraint.

**(2) Short-sighted local entropy tuning.** In each soft PIM step, as shown in Eq. (6), the existing temperature tuning mechanism aligns every local single-step entropy to a fixed target by adjusting $\alpha$ to match $\mathbb{E}[-\log \pi(a|s)]$ to some desired value. However, it would be better to adjust the trajectory entropy to control the long-term policy stochasticity, which is defined as:

$$\mathcal{H}_{\text{traj}}(s) = \mathbb{E}_{\tau \sim \pi} \Big[ \sum_{t=0}^{\infty} \gamma^t \mathcal{H}(\pi(\cdot|s_t)) \Big| s_0 = s \Big]. \tag{8}$$

In summary, while the maximum-entropy framework is a powerful tool for policy learning, its effectiveness is still hindered by the two identified bottlenecks. These limitations motivate us to execute reward-entropy separation to ensure clean and stable value learning and rethink maximum entropy RL from a trajectory-level entropy constraint perspective.

### 3.2 TRAJECTORY ENTROPY-CONSTRAINED REINFORCEMENT LEARNING

To address the two bottlenecks identified earlier, we propose Trajectory Entropy-Constrained Reinforcement Learning (TECRL). It formulates an explicit equality constraint on the trajectory-level entropy to control the policy stochasticity, which yields the following policy optimization problem:

$$\max_{\pi} \mathbb{E}_{\pi} \Big[ \sum_{t=0}^{\infty} \gamma^t [r(s_t, a_t) + \alpha \mathcal{H}(\pi(\cdot|s_t))] \Big]$$

$$\text{s.t.} \quad \mathbb{E}_{\pi} \Big[ \sum_{t=0}^{\infty} \gamma^t \mathcal{H}(\pi(\cdot|s_t)) \Big] = \mathcal{H}_{\text{budget}}. \tag{9}$$

Under this trajectory entropy constraint, the agent is required to strategically distribute a fixed budget of randomness across its entire trajectory. This offers a more principled way to mitigate the dilemma of under- and over-exploration.

To practically solve the optimal policy, our TECRL integrates four alternating steps: (1) Policy Evaluation (PEV) estimates the expected cumulative reward; (2) Policy Introspection (PIS) estimates the expected cumulative entropy; (3) Policy Improvement (PIM) jointly leverages both critics to formulate soft policy objective; and (4) Temperature Updating (TUP) adapts the temperature to enforce the trajectory entropy constraint. Below we detail these four steps one by one.

**(1) Policy Evaluation (PEV).** This step learns a reward-centric critic $Q_r$ defined as

$$Q_r(s,a) = \mathbb{E}_\pi \left[ \sum_{t=0}^{\infty} \gamma^t r_t \;\middle|\; s_0 = s, a_0 = a \right], \tag{10}$$

The PEV loss follows the standard Bellman expectation equation:

$$\mathcal{L}_{\text{PEV}} = (Q_r(s,a) - y_r)^2, \quad \text{where} \quad y_r = r(s,a) + \gamma \, \mathbb{E}_{s',a'}[Q_r(s',a')], \tag{11}$$

This reward-centric critic explicitly excludes entropy bonuses, which ensures a clean value target uninfluenced by policy stochasticity.

**(2) Policy Introspection (PIS).** This step learns an entropy-centric critic $Q_e$. For a Gaussian policy, the entropy of the current step is straightforward to compute. Therefore, we define $Q_e$ as the cumulative policy entropy from the next time step to infinity, which is defined as

$$Q_e(s,a) = \mathbb{E}_\pi \left[ \sum_{t=1}^{\infty} \gamma^t \mathcal{H}(\pi(\cdot|s_t)) \;\middle|\; s_0 = s, a_0 = a \right]. \tag{12}$$

Notably, it also does not contain the temperature $\alpha$, so its target value is clean and explicit. The PIS loss follows an entropy Bellman expectation equation:

$$\mathcal{L}_{\text{PIS}} = (Q_e(s,a) - y_e)^2, \quad \text{where} \quad y_e = \gamma \mathcal{H}(\pi(\cdot|s')) + \gamma \, Q_e(s',a'). \tag{13}$$

The mathematical correspondence between Eq. (12) and Eq. (13) can be seen in the Appendix A.2, and the convergence proof of the newly proposed PIS is presented in Appendix A.3.

We refer to this process as policy introspection because the $Q_e$ value reflects the future cumulative entropy of the current policy across different state-action pairs. In essence, it quantifies the long-term stochasticity inherent to the policy itself.

**(3) Policy Improvement (PIM).** With dual critics $Q_r$ and $Q_e$, We can formulate a policy loss as:

$$\mathcal{L}_{\text{PIM}} = \underbrace{Q_r(s,a)}_{\text{cumulative reward}} + \alpha \underbrace{(-\log \pi(a|s) + Q_e(s,a))}_{\text{cumulative entropy}}. \tag{14}$$

This PIM loss aligns with the soft policy objective shown in Eq. (1). $Q_r$ represents the cumulative reward, $-\log \pi(a|s)$ is the current policy entropy, and $Q_e$ represents the cumulative entropy starting from the next time step. Therefore, our PIM is compliant with the maximization term in Eq. (9).

**(4) Temperature Updating (TUP).** Finally, the aim of TUP is tuning $\alpha$ to enforce the trajectory entropy constraint, whose loss is

$$\mathcal{L}_{\text{TUP}} = -\alpha \left( \underbrace{-\log \pi(a|s) + Q_e(s,a)}_{\text{cumulative entropy}} - \mathcal{H}_{\text{budget}} \right). \tag{15}$$

This mechanism extends existing temperature tuning in Eq. (6) by replacing uniform local entropy matching with a trajectory-level entropy constraint in Eq. (9). We set $\mathcal{H}_{\text{budget}}$ as $\rho H_0/(1-\gamma)$, The division by $(1-\gamma)$ is to keep the magnitude consistent with the local entropy tuning of the maximum entropy. $\rho$ is an entropy scaling factor that can adjust the budget value.

**Summary.** Our proposed TECRL framework is grounded in two primary claims: (1) TECRL enables *more stable and effective exploitation*. This is because the reward-centric value function is now decoupled from the entropy objective, allowing it to provide a more accurate and dedicated prediction to guide policy improvement. (2) TECRL enables *more strategic and controllable exploration*. By having the agent dynamically allocate its finite entropy budget where it is most needed, the method facilitates the preservation of high-value behaviors while preventing unstable swings in policy stochasticity. The full pseudocode is summarized in Algorithm 1.

---

**Algorithm 1** Trajectory Entropy-Constrained Reinforcement Learning (TECRL)

---

1: **Initialize** policy $\pi_\theta$, reward critic $Q_{r,\psi}$, entropy critic $Q_{e,\phi}$, temperature $\alpha$, replay buffer $\mathcal{D}$
2: **for** each iteration **do**
3:     Observe $s_t$, sample $a_t \sim \pi_\theta(a|s_t)$, execute $a_t$, receive $r_t$, next state $s_{t+1}$
4:     Store $(s_t, a_t, r_t, s_{t+1})$ in $\mathcal{D}$
5:     Sample mini-batch $\{(s, a, r, s')\} \sim \mathcal{D}$
6:     Update $Q_r$ with Eq. (11)                          ▷ (PEV) Policy Evaluation
7:     Update $Q_e$ with Eq. (13)                          ▷ (PIS) Policy Introspection
8:     Update $\pi_\theta$ with Eq. (14)                       ▷ (PIM) Policy Improvement
9:     Update $\alpha$ with Eq. (15)                         ▷ (TUP) Temperature Updating
10: **end for**

---

### 3.3 THEORETICAL ANALYSIS ON PERFORMANCE BOUND

We formalize how a trajectory entropy constraint affects policy performance and demonstrate why a properly chosen entropy budget can raise the performance upper bound. We first denote $\pi^*_{\text{soft}}$ as the optimal policy under the standard maximum entropy RL setting, which maximizes the soft objective

$$J_{\text{MaxEnt}}(\pi^*_{\text{soft}}) = R^*_{\text{MaxEnt}} + \alpha^*_{\text{soft}} \mathcal{H}^*_{\text{soft}}, \tag{16}$$

where $R^*_{\text{MaxEnt}}$ and $\mathcal{H}^*_{\text{soft}}$ represent the optimal return and cumulative entropy, respectively, and $\alpha^*_{\text{soft}} > 0$ is the optimal temperature parameter. Let $R^*_{\text{TEC}}$ be the return upper bound of our TECRL policy. We assume that the entropy budget $\mathcal{H}_{\text{budget}}$ is chosen to be within the feasible range of entropy values encountered during the MaxEnt optimization process. Specifically, it is neither smaller than the minimal achievable entropy nor larger than the maximal entropy $\mathcal{H}^*_{\text{soft}}$ obtained by the optimal maximum-entropy policy $\pi^*_{\text{soft}}$. Therefore, with the same temperature $\alpha^*_{\text{soft}}$, we have the following inequality

$$J_{\text{MaxEnt}}(\pi^*_{\text{soft}}) \geq R^*_{\text{TEC}} + \alpha^*_{\text{soft}} \mathcal{H}^*_{\text{budget}}. \tag{17}$$

By rearranging this inequality, the return upper bound of our TECRL can be bounded from above as

$$R^*_{\text{TEC}} \leq J_{\text{MaxEnt}}(\pi^*_{\text{soft}}) - \alpha^*_{\text{soft}} \mathcal{H}_{\text{budget}}$$
$$= R^*_{\text{MaxEnt}} + \alpha^*_{\text{soft}} (\mathcal{H}^*_{\text{soft}} - \mathcal{H}_{\text{budget}}). \tag{18}$$

This inequality explicitly shows that the advantage performance bound $\Delta = R^*_{\text{TEC}} - R^*_{\text{MaxEnt}}$ is bounded by a quantity proportional to the entropy gap $\mathcal{H}^*_{\text{soft}} - \mathcal{H}_{\text{budget}}$. This analysis demonstrates that appropriately selecting $\mathcal{H}_{\text{budget}}$ can potentially lead to a higher performance bound for our TECRL.

## 4 EXPERIMENTS

### 4.1 MAIN EXPERIMENT

**Benchmark.** We evaluate performance on a suite of standard continuous control tasks from the OpenAI Gym interface (Brockman et al., 2016). Specifically, we choose 8 Mujoco tasks: Humanoid-v3, Ant-v3, Hopper-v3, Walker2d-v3, Swimmer-v3, HalfCheetah-v3, InvertedDoublePendulum-v2 (abbreviated as InvertedDP-v2) and Reacher-v2. Details are provided in Appendix B.

**Baselines.** We consider 7 well-known model-free algorithms, including trust region policy optimization (TRPO) (Schulman et al., 2015), proximal policy optimization (PPO) (Schulman et al., 2017b), deep deterministic policy gradient (DDPG) (Lillicrap et al., 2016), twin delayed deep deterministic policy gradient (TD3) (Fujimoto et al., 2018), soft actor-critic (SAC) (Haarnoja et al., 2018a), Distributional SAC (DSAC) (Duan et al., 2021) and its latest version DSAC-T (Duan et al., 2025). See Appendix D for detailed hyperparameters.

**Our method.** Our proposed DSAC-E algorithm is built on the DSAC-T, inheriting all of its hyperparameters. For the newly introduced hyperparameter $\rho$, we set its value to 20 for the Humanoid-v3 and Walker2d-v3 tasks, and to 1 for all other tasks. The reason for setting larger $\rho$ values for these two tasks is that they are relatively high-dimensional and that the robots are particularly prone to falling over due to overly random actions. Recall that the base single-step entropy budget $\mathcal{H}_0$ is a negative value, so a larger $\rho$ means a smaller budget $\rho \mathcal{H}_0/(1-\gamma)$ allocated for entropy tuning in trajectory level.

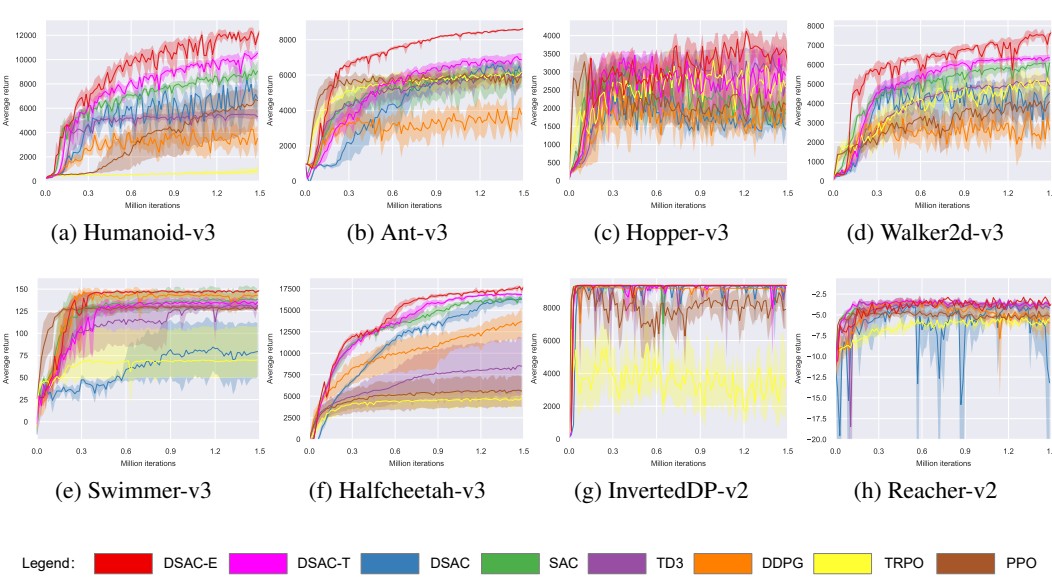

Figure 2: **Training curves on benchmarks.** The solid lines correspond to mean and shaded regions correspond to the 95% confidence interval over five runs.

Table 1: **Average final return.** Computed as the mean of the highest return values observed in the final 10% of iteration steps per run. $\pm$ corresponds to standard deviation over 5 runs.

| | Algorithm | | Humanoid-v3 | Ant-v3 | Hopper-v3 | Walker2d-v3 |
|---|---|---|---|---|---|---|
| Off | w/ entropy | DSAC-E | **12542±280** | **8640±57** | **3901±385** | **7780±137** |
| | | DSAC-T | 10829±243 | 7086±261 | 3660±533 | 6424±147 |
| | | DSAC | 9074±286 | 6862±53 | 2135±434 | 5413±865 |
| | | SAC | 9336±696 | 6427±805 | 2483±943 | 6201±264 |
| | w/o entropy | TD3 | 5632±436 | 6184±487 | 3569±455 | 5238±336 |
| | | DDPG | 5292±663 | 4549±789 | 2644±659 | 4096±68 |
| On | w/ entropy | TRPO | 965±555 | 6203±579 | 3474±400 | 5503±593 |
| | | PPO | 6869±1563 | 6157±185 | 2647±482 | 4832±638 |
| | | ⇑ | 15.82% | 21.93% | 6.58% | 21.11% |

| | Algorithm | | Swimmer-v3 | Halfcheetah-v3 | InvertedDP-v2 | Reacher-v2 |
|---|---|---|---|---|---|---|
| Off | w/ entropy | DSAC-E | **149.3±0.3** | **17904±100** | **9360±0** | **-2.9±0.1** |
| | | DSAC-T | 137.6±6.4 | 17025±157 | **9360±0** | -3.1±0.2 |
| | | DSAC | 83.9±35.6 | 16542±514 | 9359±1 | -4.3±1.9 |
| | | SAC | 140.4±14.3 | 16573±224 | **9360±0** | -3.1±0.2 |
| | w/o entropy | TD3 | 134.0±5.4 | 8633±4041 | 9347±15 | -3.4±0.2 |
| | | DDPG | 145.6±4.3 | 13970±2083 | 9183±10 | -4.5±1.3 |
| On | w/ entropy | TRPO | 70.4±38.1 | 4785±968 | 6260±2066 | -5.0±0.6 |
| | | PPO | 130.3±2.0 | 5790±2201 | 9357±2 | -4.4±0.2 |
| | | ⇑ | 2.54% | 5.16% | 0% | 6.45% |

* **Bolded** and red = best, blue = second-best. ⇑ *means the improvement of the best over the second-best.*

**Evaluation protocol.** The total training step for all experiments is set at 1.5 million, with the results of all experiments averaged over 5 random seeds. For each seed, the metric is derived by averaging the highest return values observed during the final 10% of iteration steps in each run, with evaluations

conducted every 15,000 iterations. Each assessment result is the average of ten episodes. The results from the 5 seeds are then aggregated to calculate the mean and standard deviation.

**Main results.** Figure 2 and Table 1 display all the learning curves and numerical performance results , respectively. Our comprehensive findings reveal that across all evaluated 8 tasks, the DSAC-E algorithm consistently matched or surpassed the performance of all competing benchmark algorithms, establishing new state-of-the-art results. Notably, it achieved less oscillation and substantial performance improvements on the Humanoid-v3, Ant-v3, Walker2d-v3, and Hopper-v3 tasks, with improvements of 15.82%, 21.93%, 21.11% and 6.6% over the second-best.

## 4.2 ABLATION STUDY

We conduct ablation studies on the Humanoid-v3 task to evaluate the contribution of each component.

**Reward-entropy separation (RES) and trajectory Entropy Constraint (TEC).** We perform a step-wise ablation, considering four algorithms: (1) Our full DSAC-E. (2) DSAC-E w/o TEC, which replaces our trajectory entropy constraint with existing local entropy tuning. (3) DSAC-E w/o TEC and RES, which is close to DSAC-T but with a $\rho$ value of 20. (4) original DSAC-T, which can be understood as having a $\rho$ of 1. As shown in Figure 3 and Table 2, the performance of the algorithms progressively declines as more components are removed. This result confirms the effectiveness of both our RES and TEC modules. Next we will provide a more systematic analysis of the $\rho$.

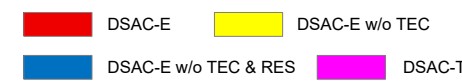

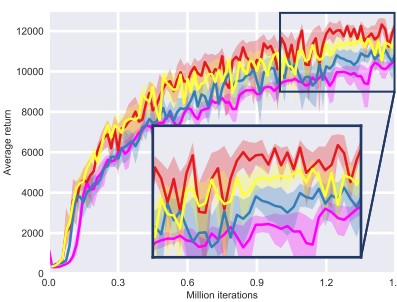

Figure 3: Ablation on the TEC and RES.

Table 2: Results of ablation on TEC and RES.

| Algorithm | TAR |
|---|---|
| DSAC-E (full) | **12542 ± 280** |
| DSAC-E w/o TEC | 11786 ± 374 |
| DSAC-E w/o TEC & RES | 11455 ± 404 |
| DSAC-T | 10829 ± 243 |

**Impact of $\rho$ controlling trajectory entropy budget.** We further investigate the effect of varying the trajectory entropy budget. Specifically, we apply different $\rho$ values for both DSAC-T and our DSAC-E. As shown in Figure 4 and Table 3, the performance gain of DSAC-T (Figure 4a) is not significant with the adjustment of $\rho$. Its performance varies only slightly and all results cluster closely together. In contrast, our DSAC-E (Figure 4b) consistently outperforms DSAC-T across all settings, and its performance shows a clearer, more structured dependence on $\rho$.

For both DSAC-T and our DSAC-E, performance first improves and then degrades as $\rho$ increases, which aligns with our theoretical analysis: a properly chosen entropy budget can lift the performance bound, whereas an excessively large $\rho$ (corresponding to an overly small entropy budget) reduces exploration and leads to a performance drop. Overall, our DSAC-E achieves higher performance and exhibits a more interpretable sensitivity to $\rho$, making it easier to tune for high returns.

Table 3: Performance of DSAC-T and our DSAC-E under different $\rho$ values.

| Algorithm | $\rho = 1$ | $\rho = 10$ | $\rho = 20$ | $\rho = 30$ |
|---|---|---|---|---|
| DSAC-T | 10829 ± 243 | 11079 ± 457 | 11455 ± 404 | 11182 ± 705 |
| DSAC-E (ours) | **11382 ± 447** | **12118 ± 505** | **12542 ± 280** | **11747 ± 365** |

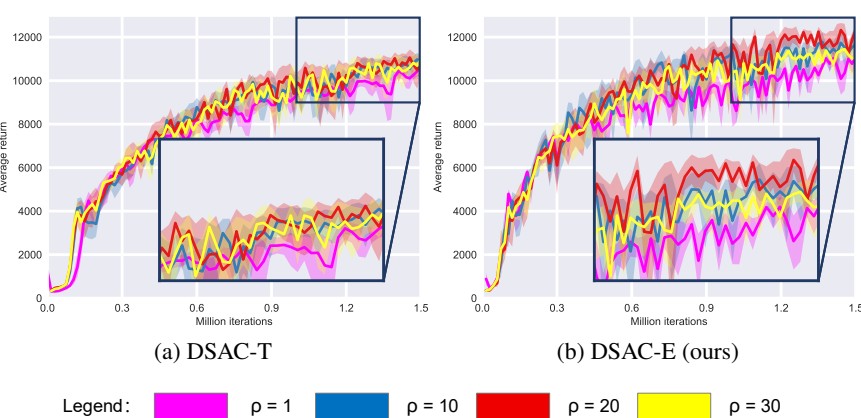

Figure 4: Ablation on the sensitivity to the trajectory entropy budget.

### 4.3 EVIDENCE FOR THE MOTIVATION

Our motivation rests on solving two "bottlenecks": (1) Non-stationary Q-value and (2) Short-sighted local tuning. We provide specific visualizations to support that these two bottlenecks do exist and impair the learning process, as shown in the Appendix F.

### 4.4 SUPPLEMENTARY RESULTS AND MORE BASELINES

We add 4 more tasks from diverse domains to strength the experimental evaluation of our method:

- **Dog-walk and Dog-run**: Two most challenging locomotion tasks in DMC.

- **Pusher**: A robotic manipulation task.

- **Carracing**: A visual-input driving task.

The detailed introduction of these environments and results is shown in the Appendix G. We also compare a new baseline $S^2AC$ (Messaoud et al., 2024), which is a recently proposed maximum entropy RL algorithm, and the results are also included.

### 4.5 GUIDANCE ON THE SELECTION OF THE HYPERPARAMETER $\rho$

Our method introduces a hyperparameter to adjust $\mathcal{H}_{\text{budget}}$. One may be concerned that this hyperparameter might be difficult to tune. Our claim is that "*Setting the default value to 1 is perfectly acceptable, and thanks to our RES, the performance is generally better than or comparable to the standard MaxEnt RL. Slightly increasing $\rho$ has a high chance of improving performance, especially for complex, high-dimensional tasks that are prone to failure due to overly random actions.*" The detailed analysis and results are shown in Appendix H.

### 4.6 TIME-EFFICIENCY OF TRAINING AND INFERENCE

For the sake of training efficiency, we employ three tasks spanning low, medium, and high dimensions (Hopper, Walker, and Humanoid) to compare two algorithms: DSAC-T and our proposed DSAC-E. All experiments are conducted on a single NVIDIA RTX 3090 Ti GPU paired with an AMD Ryzen Threadripper 3960X 24-Core Processor, using the Jax 0.4.28 programming framework. Detailed numerical results are presented in the Appendix I.

Regarding inference time-efficiency, our DSAC-E trains a same-size MLP policy as DSAC-T, so the inference time should be identical in principle.

## 5 RELATED WORK

Exploration remains a central challenge in RL, and prior studies have proposed various strategies to inject and regulate stochasticity into the policy (Amin et al., 2021). Broadly, existing approaches can be grouped into two main categories: action-noise-based and maximum-entropy-based exploration (Hao et al., 2023). While other alternatives, such as curiosity-driven (Sun et al., 2022) or uncertainty-based (An et al., 2021) exploration, have been explored, they remain less commonly adopted in standard model-free RL algorithms.

**Action-noise based exploration.** A line of methods in off-policy RL encourages exploration by directly perturbing the agent's actions with a noise process. For instance, DDPG first (Lillicrap et al., 2016) employs Ornstein–Uhlenbeck noise to facilitate temporally correlated exploration, and the TD3 family (Fujimoto et al., 2018; 2023; Seo et al., 2025) turn to simply apply Gaussian noise to each action dimension to effectively maintain randomness during training. Although these approaches are intuitive and easy to implement, they suffer from two key drawbacks. First, the noise is added externally and is entirely separate from the policy's learning objective. The policy itself is unaware of this exploration mechanism, making it a blind, ad hoc process (Plappert et al., 2018; Li et al., 2021). Second, it creates a fundamental inconsistency between training and evaluation. A policy trained with exploratory noise is different from the final policy used for deployment, which can lead to a policy-value mismatch and hinder convergence to a truly optimal solution (Hollenstein et al., 2022; Sikchi et al., 2022). Overall, although action-noise based exploration is straightforward to implement and can yield good performance, its largely heuristic nature diminishes its reliability.

**Maximum-entropy based exploration.** A more principled framework for exploration is provided by maximum-entropy RL (Haarnoja et al., 2017). By augmenting the standard RL objective with an entropy term, methods such as SAC (Haarnoja et al., 2018a) optimize for both expected return and policy entropy, thereby encouraging diverse behaviors (Nachum et al., 2017). While the latest extensions of SAC further incorporate distributional critics to improve performance (Duan et al., 2021; 2025), they share the same tuning principle of maintaining the policy's single-step entropy at a fixed target. Recent work has explored the use of generative models, such as diffusion models, as policy functions (Yang et al., 2023a; Zhu et al., 2023). While it's difficult to accurately compute the entropy of this class of functions (Yang et al., 2023b), these methods still try to follow the standard maximum-entropy principle and entropy tuning mechanism for exploration, for example, by approximating the policy entropy via GMM fitting or alternatively optimizing the lower bound (Wang et al., 2024; 2025; Ding et al., 2024; Celik et al., 2025). Their entropy tuning mechanism remains inherently uniform across all situations and does not explicitly account for long-term policy stochasticity and the inherent need for adaptive exploration. Our TECRL also employs entropy to monitor policy's stochasticity. However, we shift the focus from local entropy tuning to *trajectory entropy constraint*, highlighting a new perspective on managing policy's long-term stochasticity. We believe this work provides a new avenue for better resolving the exploitation-exploration dilemma, leading to higher performance.

## 6 CONCLUSION

In this paper, we revisit the standard maximum entropy RL framework and introduce the trajectory entropy-constrained reinforcement learning (TECRL) framework. Our work addresses two key limitations: (1) non-stationary Q-value estimation and (2) short-sighted local entropy tuning. By separating the reward and entropy Q-functions and applying the trajectory entropy constraint, our framework ensures stable value targets and effective control of long-term policy stochasticity. Building on this, we develop a practical algorithm, DSAC-E, which extends the state-of-the-art DSAC-T baseline. Empirical results on the OpenAI Gym benchmark show that DSAC-E achieves superior returns and greater stability, validating the effectiveness of our TECRL framework.

Moving forward, we plan to validate the applicability of our TECRL framework to real-world robotics and large language models (LLMs). This integration will allow agents to benefit from TECRL's superior long-term stochasticity management, leading to more effective and robust behaviors. We believe this work offers a promising paradigm for addressing the exploration-exploitation trade-off and paving the way for more powerful and robust RL agents.

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

## A  THEORETICAL ANALYSIS

### A.1  USEFUL LEMMAS

**Lemma 1** (Convergence of $\gamma$-Contraction Mappings). *Let $(X, d)$ be a complete metric space, and let $\mathcal{B} : X \to X$ be a $\gamma$-contraction mapping with $0 < \gamma < 1$. This means that for all $x, y \in X$,*

$$d(\mathcal{B}(x), \mathcal{B}(y)) \leq \gamma \cdot d(x, y), \tag{19}$$

*where $d$ is the metric on $X$. According to Banach's fixed-point theorem, $\mathcal{B}$ has a unique fixed point $x^* \in X$, such that $\mathcal{B}(x^*) = x^*$. Furthermore, for any initial point $x_0 \in X$, the iterative sequence $\{x_n\}$ defined by $x_{n+1} = \mathcal{B}(x_n)$ converges to $x^*$. The convergence rate is geometric, and we have the inequality*

$$d(x_n, x^*) \leq \gamma^n \cdot d(x_0, x^*), \quad \forall n \geq 0. \tag{20}$$

*This result not only guarantees the existence and uniqueness of the fixed point but also provides a precise rate at which the sequence approaches $x^*$, demonstrating the efficiency of contraction mappings in finding fixed points.*

### A.2  ENTROPY BELLMAN EXPECTATION EQUATION IN POLICY INTROSPECTION (PIS)

Here, we build the correspondence between the definition of $Q_e$ in Eq. (12) and the entropy Bellman expectation equation in Eq. (13).

$Q_e$ represents the cumulative policy entropy starting from the next time step, expressed as:

$$Q_e(s, a) = \mathbb{E}_\pi \left[ \sum_{t=1}^{\infty} \gamma^t \mathcal{H}(\pi(\cdot|s_t)) \,\middle|\, s_0 = s, a_0 = a \right]. \tag{21}$$

Our proposed entropy Bellman expectation equation in Eq. (13) states

$$Q_e(s, a) = \gamma \mathcal{H}(\pi(\cdot|s')) + \gamma\, Q_e(s', a'). \tag{22}$$

Substitute the definition of $Q_e$ into the RHS of Eq. (13), we have:

$$\begin{aligned}
RHS &= \gamma \mathcal{H}(\pi(\cdot|s')) + \gamma\, Q_e(s', a') \\
&= \gamma \mathcal{H}(\pi(\cdot|s_1)) + \gamma \sum_{t=1}^{\infty} \gamma^t \mathcal{H}(\pi(\cdot|s_{t+1})) \\
&= \gamma \mathcal{H}(\pi(\cdot|s_1)) + \sum_{t=1}^{\infty} \gamma^{t+1} \mathcal{H}(\pi(\cdot|s_{t+1})) \\
&= \gamma \mathcal{H}(\pi(\cdot|s_1)) + \sum_{t=2}^{\infty} \gamma^t \mathcal{H}(\pi(\cdot|s_t)) \\
&= \sum_{t=1}^{\infty} \gamma^t \mathcal{H}(\pi(\cdot|s_t)) = LHS.
\end{aligned} \tag{23}$$

Thus, we have proven that the definition of $Q_e$ is the solution of the entropy Bellman expectation equation.

### A.3  CONVERGENCE OF POLICY INTROSPECTION (PIS)

We prove the convergence of PIS by showing that the entropy Bellman operator $\mathcal{B}_e$, defined as

$$\mathcal{B}_e Q_e(s, a) = \gamma[Q_e(s', a') - \alpha \log \pi(a'|s')], \tag{24}$$

is a $\gamma$-contraction mapping.

We analyze the infinity norm of $\mathcal{B}_e$. For any two functions $Q_{e,1}(s,a)$ and $Q_{e,2}(s,a)$, we have:

$$
\begin{aligned}
\|\mathcal{B}_e[Q_{e,1}(s,a)] - \mathcal{B}_e[Q_{e,2}(s,a)]\|_\infty &= \|\gamma[Q_{e,1}(s',a') - \alpha\log\pi(a'|s')] \\
&\quad - \gamma[Q_{e,2}(s',a') - \alpha\log\pi(a'|s')]\|_\infty \\
&\leq \|\gamma Q_{e,1}(s',a') - \gamma Q_{e,2}(s',a')\|_\infty \\
&= \gamma\|Q_{e,1}(s',a') - Q_{e,2}(s',a')\|_\infty.
\end{aligned}
\tag{25}
$$

Since $\gamma \in (0,1)$, it follows that $\mathcal{B}_e$ is a $\gamma$-contraction mapping. By applying Lemma 1, we know that $\mathcal{B}_e$ has a unique fixed point. This fixed point can be obtained by iteratively applying $\mathcal{B}_e$ starting from an arbitrary initial $Q_{e,\text{init}}$. That is, as the iteration number $k$ increases, the sequence of updated Q functions converges to a fixed point, i.e., the desired $Q_e$.

## B    ENVIRONMENTAL INTRODUCTION

**MuJoCo:**   This is a high-performance physics simulation platform widely adopted for robotic reinforcement learning research. The environment features efficient physics computation, accurate dynamic system modeling, and comprehensive support for articulated robots, making it an ideal benchmark for RL algorithm development.

In this paper, we concentrate on eight tasks: Humanoid-v3, Ant-v3, HalfCheetah-v3, Walker2d-v3, InvertedDoublePendulum-v3 (InvertedDP-v2), Hopper-v3, Reacher-v2, and Swimmer-v3, as illustrated in Figure 5. The InvertedDP-v3 task entails maintaining the balance of a double pendulum in an inverted state. In contrast, the objective of the other tasks is to maximize the forward velocity while avoiding falling. All these tasks are realized through the OpenAI Gym interface (Brockman et al., 2016).

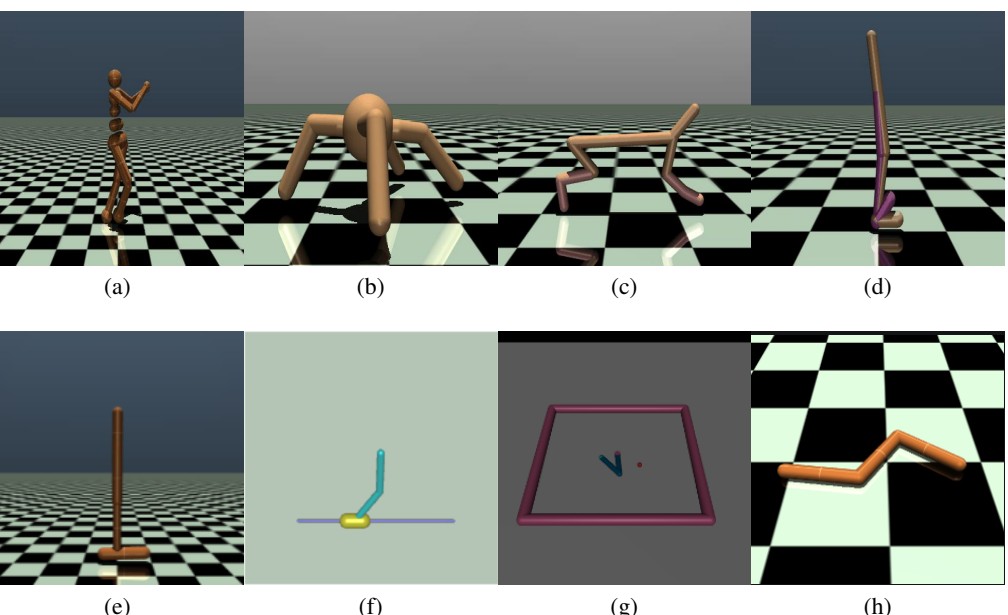

Figure 5: Benchmarks. (a) Humanoid-v3: $(s \times a) \in \mathbb{R}^{376} \times \mathbb{R}^{17}$. (b) Ant-v3: $(s \times a) \in \mathbb{R}^{111} \times \mathbb{R}^{8}$. (c) HalfCheetah-v3: $(s \times a) \in \mathbb{R}^{17} \times \mathbb{R}^{6}$. (d) Walker2d-v3: $(s \times a) \in \mathbb{R}^{17} \times \mathbb{R}^{6}$. (e) Hopper-v3: $(s \times a) \in \mathbb{R}^{11} \times \mathbb{R}^{3}$. (f) InvertedDoublePendulum-v2: $(s \times a) \in \mathbb{R}^{6} \times \mathbb{R}^{1}$. (g) Reacher-v2: $(s \times a) \in \mathbb{R}^{11} \times \mathbb{R}^{2}$. (h) Swimmer-v3: $(s \times a) \in \mathbb{R}^{8} \times \mathbb{R}^{2}$.

## C  VISUALIZATIONS

To demonstrate the effectiveness of DSAC-E in solving complex, high-dimensional locomotion tasks, we provide visualizations of policy control process on three of the most challenging benchmarks in the Humanoid task as shown in the following Figure 6. These tasks require precise coordination across many degrees of freedom and long-horizon reasoning.

The visualization showcase that DSAC-E not only achieves successfully running but also learns robust posture and behaviors, highlighting its strong capabilities in difficult control scenarios.

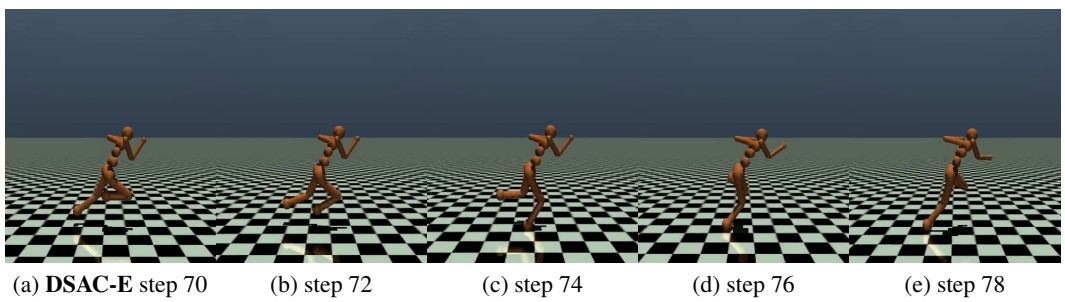

(a) **DSAC-E** step 70  (b) step 72  (c) step 74  (d) step 76  (e) step 78

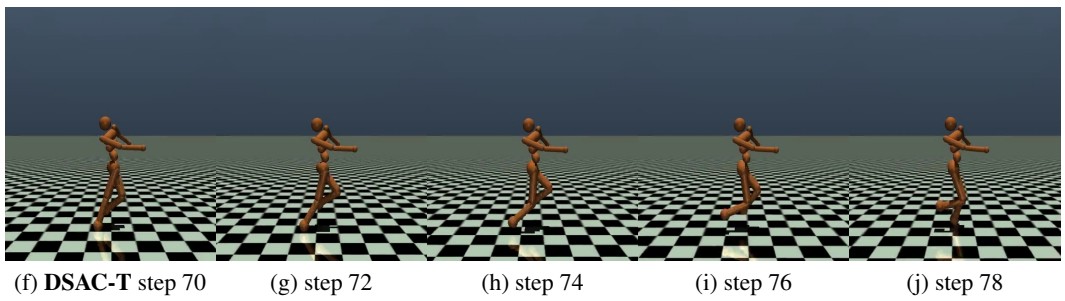

(f) **DSAC-T** step 70  (g) step 72  (h) step 74  (i) step 76  (j) step 78

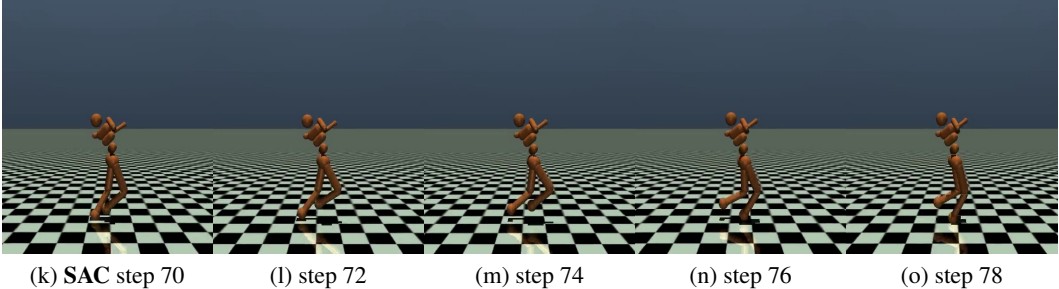

(k) **SAC** step 70  (l) step 72  (m) step 74  (n) step 76  (o) step 78

Figure 6: Visualizations of control processes on Humanoid-v3 task.

## D  REPRODUCIBILITY STATEMENT

TABLE 4
DETAILED HYPERPARAMETERS.

| Hyperparameters | Value |
|---|---|
| *Shared* | |
| Optimizer | Adam ($\beta_1 = 0.9, \beta_2 = 0.999$) |
| Actor learning rate | 1e−4 |
| Critic learning rate | 1e−4 |
| Discount factor ($\gamma$) | 0.99 |
| Policy update interval | 2 |
| Target smoothing coefficient ($\tau$) | 0.005 |
| Reward scale | 0.1 |
| Number of iterations | $1.5 \times 10^6$ |
| *Maximum-entropy framework* | |
| Learning rate of temperature $\alpha$ | $3 \times 10^{-4}$ |
| Base expected entropy ($\overline{\mathcal{H}}$) | $\overline{\mathcal{H}} = -\dim(\mathcal{A})$ |
| *Deterministic policy* | |
| Exploration noise | $\epsilon \sim \mathcal{N}(0, 0.1^2)$ |
| *Off-policy* | |
| Sample batch size | 20 |
| Replay batch size | 256 |
| Replay buffer warm size | $1 \times 10^4$ |
| Replay buffer size | $1 \times 10^6$ |
| *On-policy* | |
| Sample batch size | 2000 |
| Replay batch size | 2000 |
| GAE factor | 0.95 |
| *DSAC-T* | |
| Variance clipping constant $\zeta$ | 3 |
| Stabilizing constant $\epsilon$ and $\epsilon_\omega$ | 0.1 |
| *DSAC-E (ours)* | |
| $\rho$ | 20 for Humanoid and Walker2d, otherwise 1 |

## E  LLM USAGE DISCLOSURE

We used ChatGPT to polish grammar and improve text clarity. We reviewed all LLM-generated suggestions and are fully responsible for the final content of this paper.

# F EVIDENCE FOR MOTIVATION

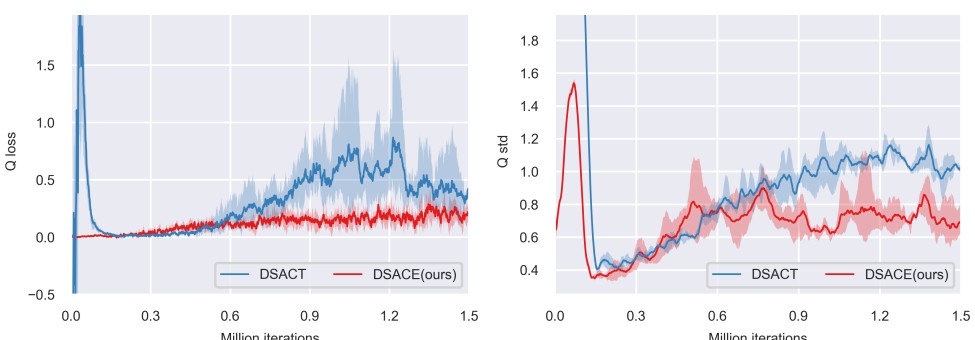

Figure 7: Evidence for non-stationary Q-value.

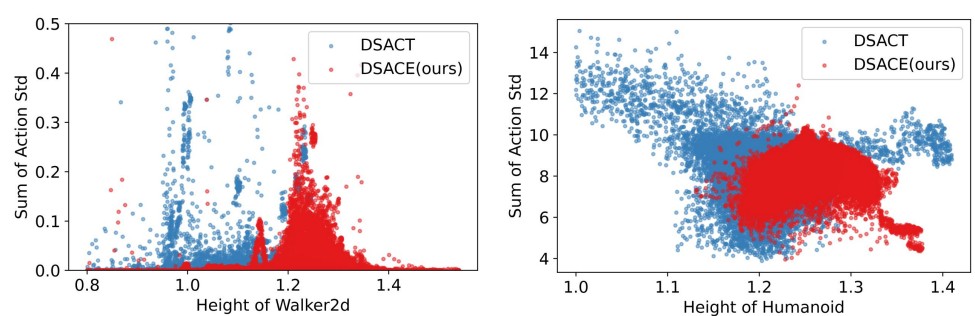

Figure 8: Evidence for short-sighted local tuning.

# G SUPPLEMENTARY RESULTS AND MORE BASELINES

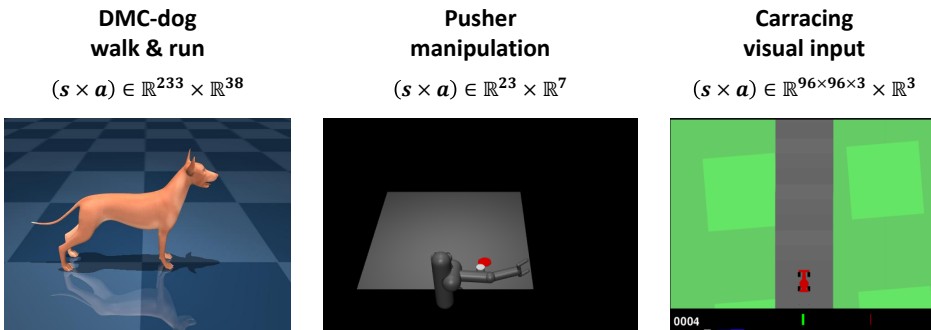

Figure 9: Snapshots of the additional tasks.

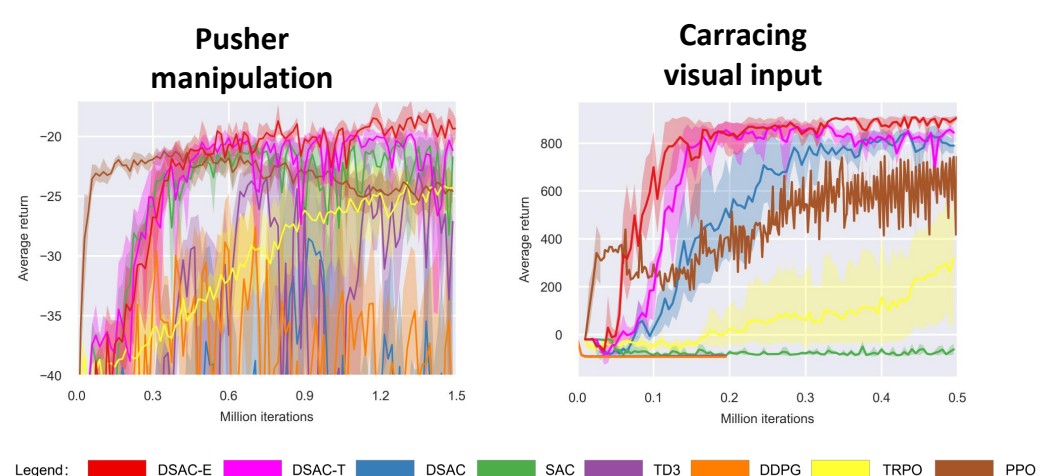

Figure 10: Results on the pusher and carracing tasks.

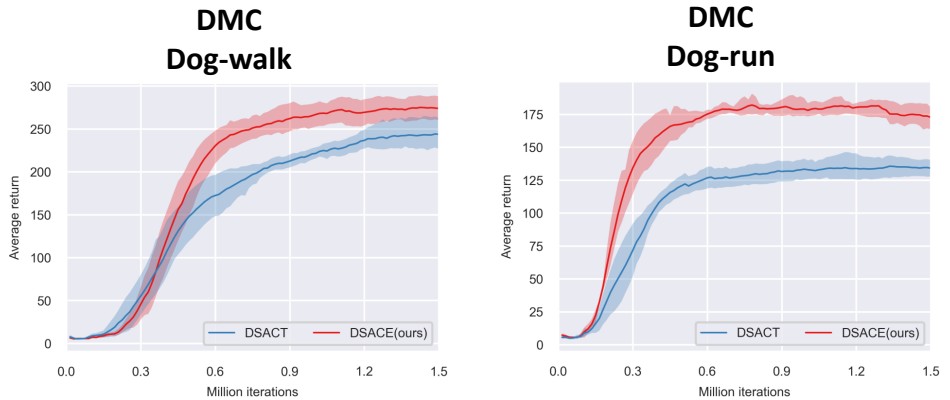

Figure 11: Results on the dmc dog-walk and dog-run tasks.

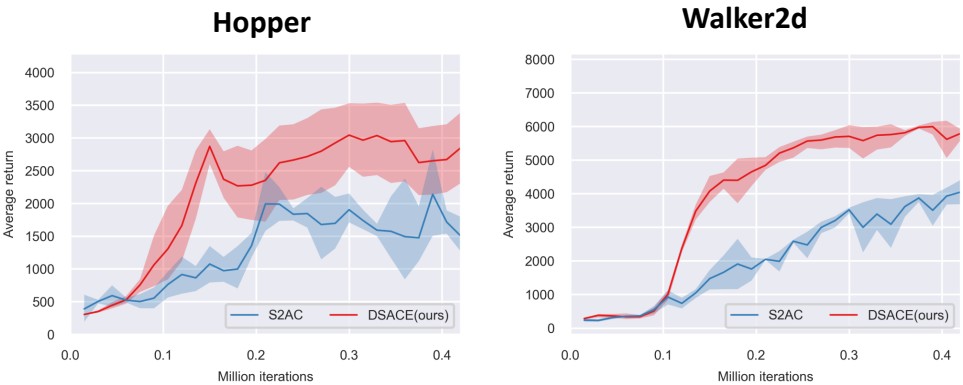

Figure 12: Comparison with $S^2AC$.

## H GUIDANCE ON THE SELECTION OF THE HYPERPARAMETER $\rho$

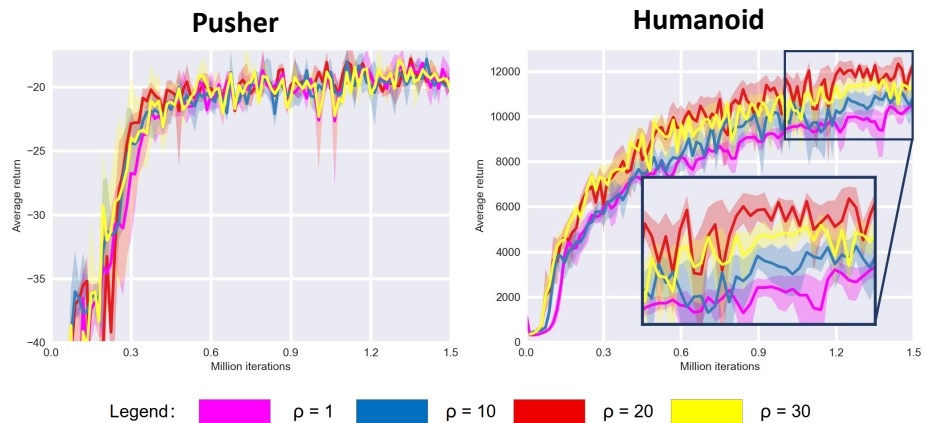

Figure 13: Ablation on the impact of $\rho$ values.

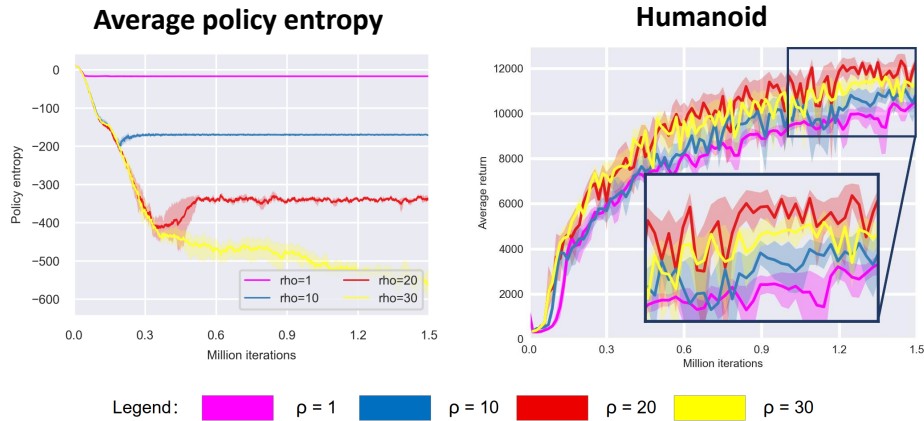

Figure 14: Visualization of the policy entropy controlled by $\rho$ values.

## I TIME-EFFICIENCY OF TRAINING AND INFERENCE

| Algorithm | Humanoid (376,17) | Ant (111,8) | Walker2d (17,6) |
|---|---|---|---|
| **DSAC-T** | 2h09m (129) | 1h47m (107) | 1h29m (89) |
| **DSAC-E** | 2h23m (143) | 2h02m (122) | 1h44m (104) |
| **Percentage** | **110.8%** | **114.0%** | **116.9%** |

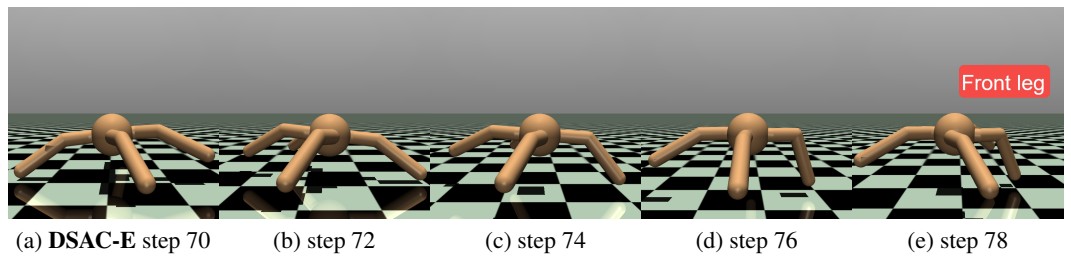

(a) **DSAC-E** step 70    (b) step 72    (c) step 74    (d) step 76    (e) step 78

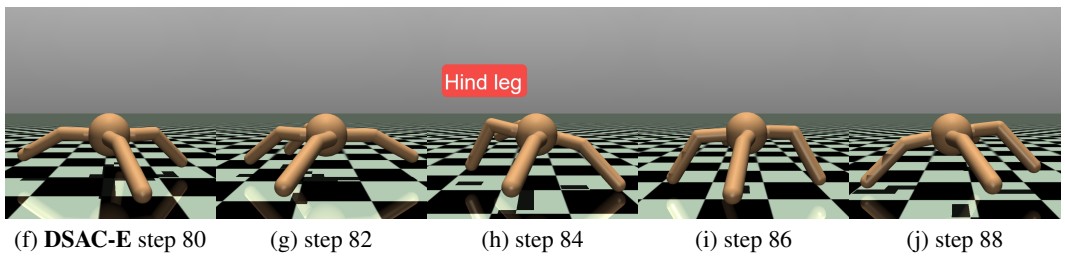

(f) **DSAC-E** step 80    (g) step 82    (h) step 84    (i) step 86    (j) step 88

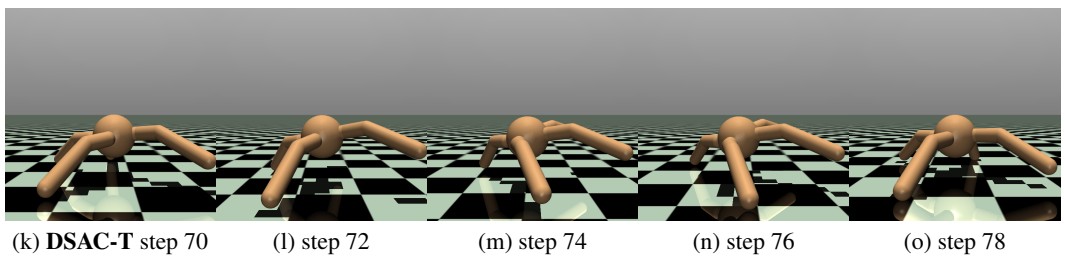

(k) **DSAC-T** step 70    (l) step 72    (m) step 74    (n) step 76    (o) step 78

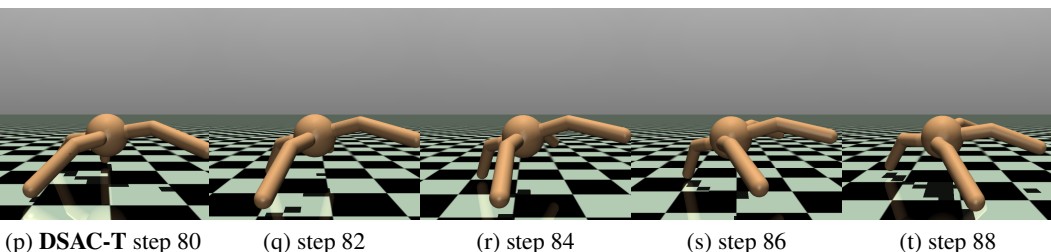

(p) **DSAC-T** step 80    (q) step 82    (r) step 84    (s) step 86    (t) step 88

Figure 15: Visualizations of control processes on Ant task.

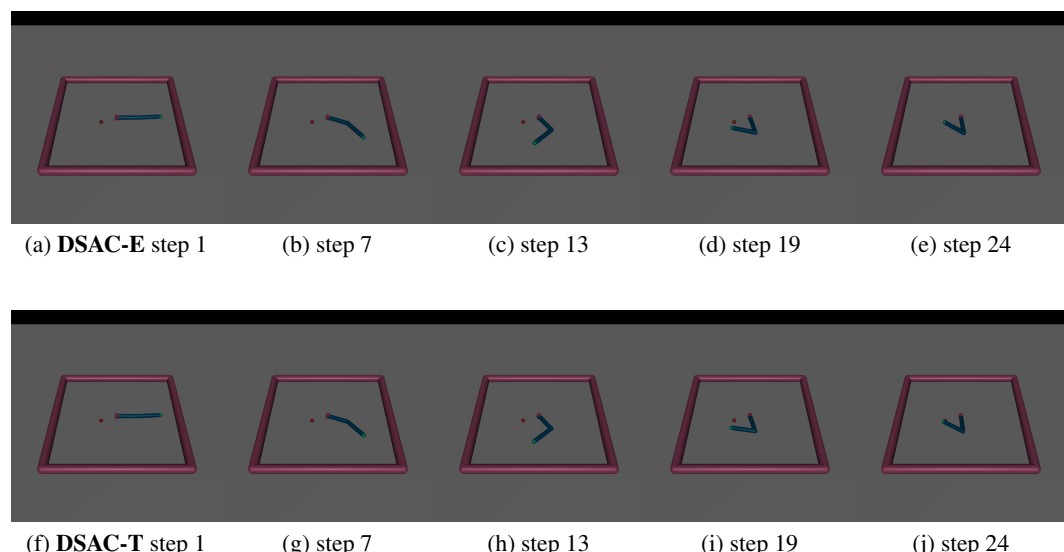

(a) **DSAC-E** step 1  (b) step 7  (c) step 13  (d) step 19  (e) step 24

(f) **DSAC-T** step 1  (g) step 7  (h) step 13  (i) step 19  (j) step 24

Figure 16: Visualizations of control processes on Reacher task.

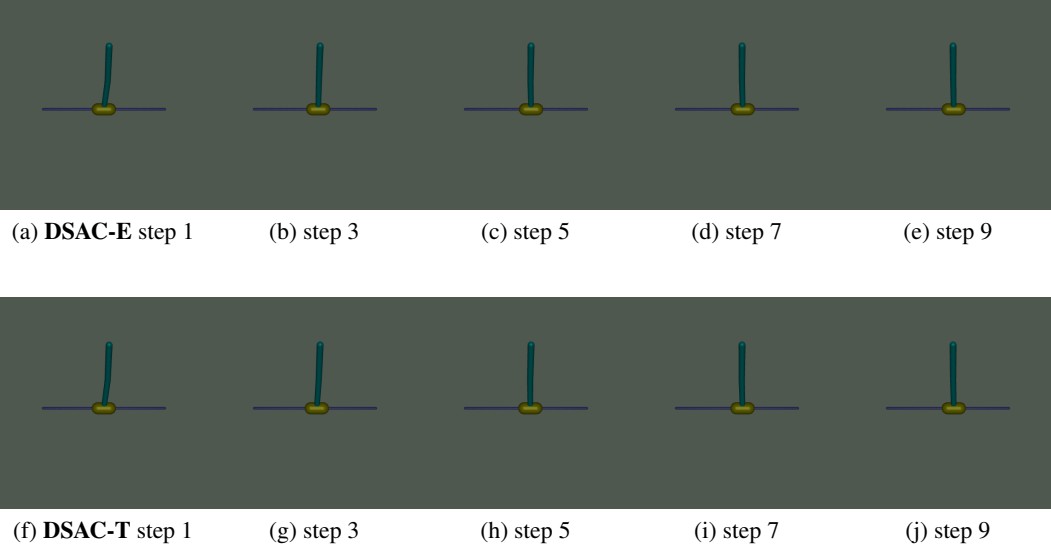

(a) **DSAC-E** step 1  (b) step 3  (c) step 5  (d) step 7  (e) step 9

(f) **DSAC-T** step 1  (g) step 3  (h) step 5  (i) step 7  (j) step 9

Figure 17: Visualizations of control processes on IDP task.

**Multi-Goal Evaluation During Training**

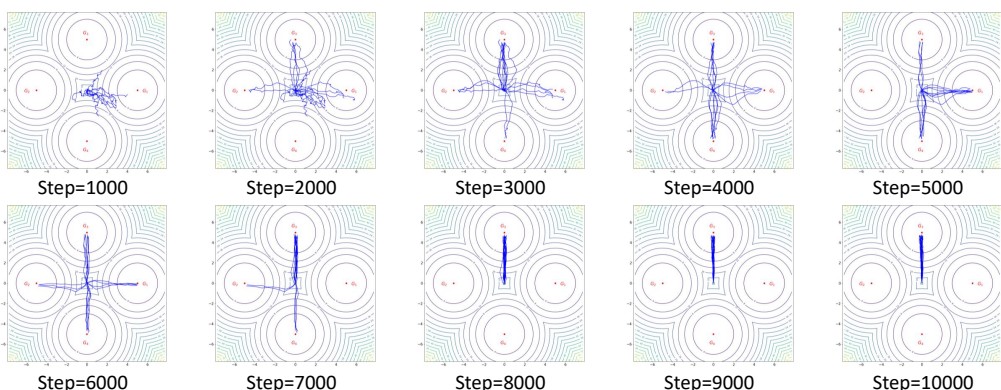

Figure 18: Visualizations of control processes on Multi-goal task.

