# OpenReview forum: "Mind Your Entropy: From Maximum Entropy to Trajectory Entropy-Constrained RL"
_ICLR.cc/2026/Conference — Submitted to ICLR 2026_

### Official Review · Reviewer_H9bY · 2025-10-19

**Soundness:** 2
**Presentation:** 2
**Contribution:** 2
**Rating:** 2
**Confidence:** 3

**Summary:**

This paper proposes Trajectory Entropy-Constrained Reinforcement Learning (TECRL), which augments maximum-entropy RL by introducing a separate entropy Q-function to estimate cumulative (trajectory-level) entropy and enforce an entropy constraint, thereby decoupling reward and entropy learning to mitigate non-stationarity from temperature updates. Built upon the DSAC framework, the method (DSAC-E) is evaluated on continuous-control benchmarks where it shows moderate improvements in stability and return.

**Strengths:**

(1) The method is motivated by a clear limitation in standard maximum-entropy RL: non-stationary targets when temperature is updated.

(2) The trajectory-entropy concept is intuitively appealing and could inspire alternative formulations of long-term exploration control.

(3) The proposal is practical and seems to fit within existing off‐policy actor‐critic frameworks, which aids reproducibility and relevance to applied RL.

(4) Empirical results appear to show meaningful improvement (both higher returns + improved stability) on commonly used benchmarks, which strengthens the claim of benefit.

**Weaknesses:**

(1) Omission of relevant baseline:
S²AC[1], a max-entropy RL method results in maximizing the expected future entropy without the need for tuning the temperature parameter (Figure 2). The policy is modeled as SVGD sampler. A sufficient number of steps leads to good convergence in bothe smooth and non-smooth landscapes.

(2) Evaluating the proposed method on the multi-goal environment used in S²AC [1] would strengthen the paper’s claims and help illustrate the intuition behind trajectory-level entropy control.

(3)  The contribution appears incremental relative to existing entropy-regularized RL frameworks.

(4) The write-up lacks some refinement: The text repeatedly refers to “non-stationary Q-values,” whereas the non-stationarity arises in the distribution induced by joint reward–entropy learning, not in the Q-values themselves. Also, the effect of the tempreture parameter on the actor stationarity and the short-sighted local entropy tuning only becomes clear in section 2, the introduction need to be adjusted to foster early understanding of these key concepts.


[1] Messaoud S, Mokeddem B, Xue Z, Pang L, An B, Chen H, Chawla S. S $^ 2$ AC: Energy-Based Reinforcement Learning with Stein Soft Actor Critic. ICLR, 2024.

**Questions:**

(1) How does your approach compare S²AC[1]?
(2) Could you evaluate your method on the multi-goal environment used in S²AC [1]?

---

> ### Author Response · Authors · 2025-11-21
>
> We sincerely appreciate your detailed and constructive feedback. Below are our responses to your comments.
>
> ## W1&Q1: Clarifying the Relationship Between S²AC and TECRL
> > Omission of relevant baseline: S²AC[1], a max-entropy RL method results in maximizing the expected future entropy without the need for tuning the temperature parameter (Figure 2). The policy is modeled as SVGD sampler. A sufficient number of steps leads to good convergence in both smooth and non-smooth landscapes.
>
> > How does your approach compare S²AC？
>
> We sincerely thank you for highlighting the need to compare our TECRL with S²AC. We would like to clarify that S²AC and TECRL address orthogonal research problems within the MaxEnt RL framework, and therefore provide complementary advancements rather than competing contributions.
>
> ### 1.1 What S²AC contributes:
> - **EBM-based policy representation:** S²AC introduces an EBM-based policy, which breaks free from the Gaussian prior that constrains most actor–critic algorithms. The EBM representation can express arbitrarily complex, highly multi-modal action distributions, which Gaussian policies fundamentally cannot capture. This allows S²AC to effectively explore and exploit environments with multi-peak return landscapes, as demonstrated in their multi-goal experiments.
>
> ### 1.2 What TECRL contributes:
> - **Accurate critic learning (via RES):** TECRL separates reward and entropy learning via reward–entropy decoupling, resulting in stable value learning and consistent Boltzmann trajectory weighting.
> - **Adaptive entropy allocation mechanism (via TEC):** TECRL introduces trajectory-level entropy regularization, enabling temporally adaptive exploration and solving MaxEnt RL’s structural short-sighted limitations in entropy control.
>
> Thus, while S²AC improves policy representation (It models the policy as a Stein Variational Gradient Descent (SVGD) sampler from an EBM over Q-values), TECRL improves Q-value accuracy and proposes a more effective entropy tuning mechanism. These two are methodologically orthogonal.
>
> We find that S²AC still adopts the standard MaxEnt RL framework with coupled reward–entropy learning and per-step local entropy tuning, and does not use automatic temperature tuning (Please see the Lines 91, 138, and 146 in https://github.com/SafaMessaoud/S2AC-Energy-Based-RL-with-Stein-Soft-Actor-Critic/blob/07b7e7c7e7816f37e670e498ffbbdc145a364893/STAC/core.py)
>
>  Given their orthogonality, combining TECRL (ours) with SVGD/EBM-based policy families (proposed by S²AC) is a highly promising direction.
>
> ## Q2: Empirical comparison
> > Evaluating the proposed method on the multi-goal environment used in S²AC would strengthen the paper’s claims and help illustrate the intuition behind trajectory-level entropy control.
> > Could you evaluate your method on the multi-goal environment used in S²AC?
>
> Our experimental results further validate this orthogonality:
> - In continuous-control locomotion tasks (e.g., Hopper, Walker) shown in [[Figure11_results_compare_s2ac]](https://anonymous.4open.science/r/ICLR2026_DSACE_FIG-BBD7/Figure11_results_compare_s2ac.pdf) (Figure 12 in the revised paper), TECRL/DSACE outperforms S²AC significantly. This is because such environments benefit more from accurate Q-value learning and adaptive entropy allocation, which are the core strengths of TECRL.
>
> - In multi-goal environment, the generative policy S²AC can reach multiple peaks in the return landscape, whereas our Gaussian policy, illustrated in [[Figure12_viz_multi_goal]](https://anonymous.4open.science/r/ICLR2026_DSACE_FIG-BBD7/Figure12_viz_multi_goal.pdf) (Figure 18 in the revised paper), can only reach one peak. This difference is expected and consistent with the theoretical design of both methods. **The lack of multi-modal representation is an inherent limitation of Gaussian policies. Our contribution lies in advancing the algorithmic updating design of Gaussian policies by enhancing the Q-value estimation accuracy and optimizing the entropy tuning mechanism.**
>
> Combining TECRL with SVGD/EBM-based policy families offers a promising direction, as SVGD can generate multi-modal policies while TECRL enables high-accuracy Q-value estimation and trajectory-level exploration allocation. This synergy allows the policy to benefit from both structured exploration and multi-modal nature, enhancing performance across complex tasks requiring diverse strategies.

---

> ### Author Response · Authors · 2025-11-21
>
> ## W3: Detailed novelty analysis of TECRL
> > The contribution appears incremental relative to existing entropy-regularized RL frameworks.
>
> Thanks for your feedback. We respectfully provide a more detailed clarification in response to your concerns. While TECRL is rooted in the MaxEnt RL framework, it is not a minor incremental extension. Instead, it addresses two under-addressed fundamental bottlenecks and effectively extends MaxEnt RL by introducing two fundamentally new mechanisms (RES and TEC) for obtaining much improved performance.
>
> - **Reward-entropy separation (RES) solves non-stationary Q-value estimation (Bottleneck 1):** TECRL separates reward and entropy learning via reward–entropy decoupling, resulting in stable and accurate value learning. Please see [Figure01_Q_evidence](https://anonymous.4open.science/r/ICLR2026_DSACE_FIG-BBD7/Figure01_evidence.pdf) (Figure 7 in the revised paper), where we provide direct empirical evidence via Q-loss and Q-value standard deviation (std) curves (comparing DSAC-T and our DSAC-E) on the Humanoid task.
>
> - **Trajectory entropy constraint (TEC) solves short-sighted local entropy tuning (Bottleneck 2):** TECRL introduces trajectory-level entropy constraint to tune temperature, enabling temporally adaptive exploration. Please see  [Figure02_entropy_evidence](https://anonymous.4open.science/r/ICLR2026_DSACE_FIG-BBD7/Figure02_entropy_evidence.pdf) (Figure 8 in the revised paper), where we visualize policy entropy distribution across the state space. our TECRL enables strategic entropy allocation: preserving exploration in normal states while reducing randomness in high-risk regions to avoid termination.
>
> In summary, the TECRL enjoys more accurate Q-value learning and enables adaptive exploration allocation in accordance with the characteristic of the task. Thus, TECRL achieves significantly higher asymptotic performance, as confirmed by our empirical results.
>
> ## W4: Refine the write-up
> > The write-up lacks some refinement: The text repeatedly refers to "non-stationary Q-values," whereas the non-stationarity arises in the distribution induced by joint reward–entropy learning, not in the Q-values themselves. Also, the effect of the temperature parameter on the actor stationarity and the short-sighted local entropy tuning only becomes clear in section 2, the introduction need to be adjusted to foster early understanding of these key concepts.
>
> We thank you for the insightful comments and agree that some of our original statements may have unintentionally caused confusion, hindering early understanding of key concepts. We provide clarifications and have revised the relevant statements as follows:
>
> - **On the non-stationary Q-values**
>   Our intention was to highlight that the reward–entropy coupling leads to non-stationary learning targets, because the temperature parameter and policy entropy continuously reshape the soft Bellman target. We acknowledge the bootstrapping mechanism of Bellman equation is also a source of non-stationarity. We have added more explanations in the introduction to distinguish this point:
>
>   "*We acknowledge that the bootstrapping update mechanism of Q-values contributes significantly to the non-stationarity in RL. In this context, we highlight that the coupling of reward and entropy is another crucial contributing factor, and our method can effectively address and eliminate this factor.*"
>
> - **On the short-sighted local entropy tuning**
>   We acknowledge your concern that the introduction does not sufficiently demonstrate the mechanism of short-sighted local entropy tuning and how actor updating is affected. We have added more explanations in the introduction to introduce it clearly:
>
>   "*While some works have explored constraining entropy (increase the temperature if the entropy at the current step falls below a target value, and decrease it otherwise), they all suffer from short-sighted local entropy tuning.*"
>
>   "*More critically, why we say they are short-sighted is that they enforce a uniform entropy target for each current state, as if every situation demands the same degree of randomness. This one-size-fits-all assumption is overly restrictive and fails to account for the inherent variability in the dynamics of different states. Consequently, the actor update process is compromised, as it neglects the fact that effective exploration should take into consideration both the underlying system dynamics and the agent’s learning progress*"
>
> ## Final thanks
> Thank you again for your time, effort, and professionalism. We hope our responses address your questions and concerns. We are happy to provide additional details if needed, and we look forward to further discussion.

---

> > ### Comment · Reviewer_z3ry · 2025-11-24
> >
> > Hi, I’m Reviewer z3ry. I read the other reviewers’ comments and feel that one of your reasons for recommending rejection may be based on a misinterpretation.
> >
> > Regarding the cited work [1], I believe TECRL is completely orthogonal to this study. The proposed approach is general and can be applied to any off-policy RL method. In contrast, S2AC [1] is a specific extension, and this work has already demonstrated that the method can be successfully applied to the state-of-the-art off-policy RL framework [2]. Therefore, I don’t think it is necessary to additionally apply it to S2AC.
> >
> > ---
> >
> > [2] Duan et al. Distributional soft actor-critic: Off-policy reinforcement learning for addressing value estimation errors. TNNLS 2021.

---

> > ### Author Response · Authors · 2025-11-27
> > **Kind Reminder**
> >
> > Thank you again for the great efforts and valuable comments. We have dedicated significant effort to addressing your main concerns. Specifically regarding S2AC, as you suggested, we have added extensive comparative experiments and a detailed discussion to highlight the orthogonality between the two approaches (with proper citations). Besides, we are pleased to share that Reviewer z3ry also agrees with our analysis on this point.
> >
> > With the discussion window closing soon, we respectfully hope you might find our response satisfactory. We look forward to hearing from you and are happy to clarify any final questions.

---

### Official Review · Reviewer_Z1ff · 2025-10-23

**Soundness:** 3
**Presentation:** 2
**Contribution:** 3
**Rating:** 4
**Confidence:** 4

**Summary:**

The paper "Mind Your Entropy: From Maximum Entropy to Trajectory Entropy-constrained RL" identifies two bottlenecks in standard maximum-entropy RL and proposes a trajectory entropy-constrained RL framework to address them. The approach decouples the critic into a reward-centric and an entropy-centric critic to avoid non-stationary Q-value estimation caused by automatic temperature adjustment. The entropy critic further enables a trajectory-level definition of entropy. Experiments on several MuJoCo tasks show performance gains over baseline algorithms.

**Strengths:**

1. The idea of separating the critic into reward-centric and entropy-centric components is intuitive and conceptually clear.
2. The trajectory-level entropy constraint provides a novel perspective that could inspire future work on entropy-based control.
3. The empirical results show noticeable improvements on several MuJoCo environments compared to strong baselines.

**Weaknesses:**

- Unclear theoretical analysis:

The analysis in Subsection 3.3 is not sufficiently clear. It is not convincing that the trajectory entropy budget necessarily leads to a higher performance bound. The metric used to define the "performance bound" should be explicitly stated. Equation (18) alone does not imply that enforcing an entropy budget guarantees performance improvement (the inequality logic C <= A+B does not lead to C >= A).

- Inconsistent experimental results:

Some results differ from those reported in the original DSAC-T paper (e.g., Reacher-v2, where the original reports -3+/-0). The paper should clarify the source of these discrepancies and justify why only eight environments are selected for evaluation.

- Ambiguity in Equation (9):

The definition of H_budget in Eq. (9) is unclear. Please provide a precise explanation or derivation.

- Unexplained negative entropy budget:

Section 4.1 (“Our method”) mentions that $H_\theta$ is a negative value. The reasoning behind this choice should be discussed.

**Questions:**

Questions are already discussed in the Weaknesses section, but are listed below for clarity:

- In Subsection 3.3, how exactly does the trajectory entropy budget lead to a higher performance bound? Please clarify the theoretical connection and specify the metric defining "performance bound".

- In Eq. (9), how is 𝐻_budget formally defined or derived?

- Why is ​$H_\theta$ stated to be negative in Sec. 4.1?

- Some experimental results (e.g., Reacher-v2) differ from the DSAC-T paper. Could you explain the source of these discrepancies and the reason for evaluating only eight environments?

---

> ### Author Response · Authors · 2025-11-21
>
> We sincerely appreciate your detailed and constructive feedback. Below are our responses to your comments.
>
> ## W1&Q1: Clarification on performance bound and entropy budget
> > In Subsection 3.3, how exactly does the trajectory entropy budget lead to a higher performance bound? Please clarify the theoretical connection and specify the metric defining "performance bound".
>
> We thank you for pointing out that the presentation in Section 3.3 was not sufficiently clear. We agree that the term "performance bound" and the inequality following Eq. (18) require a more explicit explanation. We clarify these points below.
>
> ### 1.1 Notation
>  In our analysis, the "performance bound" refers specifically to the upper bound of the discounted sum of rewards. Following your simple notation, we denote:
> - $A$: the performance upper bound of the standard MaxEnt RL.
> - $C$: the performance upper bound of our TECRL.
> - $\Delta = C - A$: the performance-bound advantage of TECRL relative to MaxEnt RL
>
> ### 1.2 Clarifying inequality and its implications
> We acknowledge that inequality of the form $C \le A + B$ does not imply $C \ge A$. However, our intention was not to claim such a direct implication. Instead, our result is that the difference between the two bounds satisfies: $\Delta = C - A \le B,$ where $B\propto (\mathcal{H}^\star - \mathcal{H}_{\text{budget}})$. And the  quantity $B$ measures the additional flexibility provided by the trajectory-entropy budget. Thus, our theoretical result does not claim that TECRL always has a strictly higher bound than MaxEnt RL. Rather, it shows that: adjusting the trajectory entropy budget $\mathcal{H}\_{\text{budget}}$ (via $\rho$) can increase $B$, which upper-bounds the potential performance-bound advantage $\Delta$. This aligns with our empirical observation that performance first improves and then degrades as $\rho$ increases for the Humanoid task.
>
> ## W2&Q2: Definition of $\mathcal{H}\_{\text{budget}}$
> > In Eq. (9), how is $\mathcal{H}\_{\text{budget}}$ formally defined or derived?
>
> Thank you for raising this important question. We provide a detailed explanation of the trajectory entropy budget $\mathcal{H}\_{\text{budget}}$ below.
>
> In MaxEnt RL, the temperature tuning mechanism maintains the single-step policy entropy close to a predefined target $\mathcal{H}\_0 = -\mathrm{dim}(\mathcal{A})$, which regulates the desired level of local stochasticity. In contrast, TECRL constrains the long-horizon discounted trajectory entropy: $\mathcal{H}\_{\text{traj}}= \sum\_{t=0}^{\infty}\gamma^t \mathcal{H}(\pi(\cdot \mid s\_t))$.
>
> To ensure **magnitude consistency** with the maximum-entropy framework, the default $\mathcal{H}\_{\text{budget}}$ in Eq. (9) is defined as the discounted accumulation of single-step entropy target $\mathcal{H}\_0$. If every step the policy entropy achieves entropy $\mathcal{H}\_0$, then: $\mathcal{H}\_{\text{traj}}= \sum_{t=0}^{\infty} \gamma^t \mathcal{H}\_0= \frac{\mathcal{H}\_0}{1-\gamma}$. Therefore, we define the trajectory entropy budget as $\mathcal{H}\_{\text{budget}} = \frac{\rho\, \mathcal{H}\_0}{1-\gamma},$ where $\rho=1$ is the default setting and it can adjust the level of long-horizon stochasticity.

---

> > ### Comment · Reviewer_Z1ff · 2025-11-25
> >
> > Thanks for the authors' responses. For the "1.2 Clarifying inequality and its implications" section, I am still not fully convinced. Specifically, we cannot guarantee that "B" itself is the lowest boundary for $\Delta$. Let's say $\Delta$ $\leq Z \leq$ $B_1$ $\leq$ $B_2$. Adjust $\rho$ (I believe is to reduce $\rho$) can push $B_1$ to $B_2$, this does not change the fact that $\Delta$ $\leq Z$. Please correct me if I am wrong.

---

> > > ### Author Response · Authors · 2025-11-26
> > >
> > > We thank you for this insightful comment. We would like to clarify the mathematical definition and then address the logical implications of the bound.
> > >
> > > ## 1. Clarification on $\rho$ and $B$:
> > >
> > > First, we wish to clarify the relationship between $\rho$ and the bound. Since $\mathcal{H}\_{\text{budget}} = \frac{\rho\mathcal{H}\_0}{1-\gamma} < 0$, increasing $\rho$ makes $\mathcal{H}\_{\text{budget}}$ **more negative (larger magnitude)**. Consequently, since $B \propto (\mathcal{H}^\star - \mathcal{H}\_{\text{budget}})$, **increasing $\rho$ actually increases the upper bound $B$.**
> > >
> > > ## 2. On the Inequality $\Delta \le Z \le B$:
> > >
> > > We agree with your reasoning that a tighter, intrinsic bound $Z$ might exist ($\Delta \le Z \le B$), and simply raising $B$ does not strictly guarantee an increase in $\Delta$. **However, increasing $B$ acts as a necessary condition for a larger advantage.** If $B$ is too small (e.g., $B < Z$), it becomes the active bottleneck limiting $\Delta$. Therefore, our claim is that adjusting $\rho$ allows us to expand $B$ so that it is no longer the limiting factor, thereby admitting the possibility of a larger $\Delta$. This is why we carefully phrase it as *"increasing $B$... upper-bounds the potential performance-bound advantage."* We welcome any suggestions on phrasing to represent this relationship more accurately.
> > >
> > > ## 3. Empirical Verification:
> > >
> > > Our empirical results corroborate this view. To demonstrate the practical effect of loosening this bound, **we conducted additional ablation studies on $\rho$ during the rebuttal phase**. The results are shown in [Figure06_ablation_on_rho](https://anonymous.4open.science/r/ICLR2026_DSACE_FIG-BBD7/Figure06_ablation_on_rho.pdf) and the numerical results are listed in the following Table:
> > >
> > > |       | $\rho$ = 1         | $\rho$ = 10        | $\rho$ = 20        | $\rho$ = 30        |
> > > |----------------|----------------|----------------|----------------|----------------|
> > > | **Humanoid** | 11382 ± 447  | 12118 ± 505 | 12542 ± 280 | 11747 ± 365 |
> > > | **Pusher**   | -17.7 ± 0.2  |-17.6 ± 0.4  |-17.6 ± 0.3  |-18.0 ± 0.5  |
> > >
> > > These results suggest that $\rho$ is robust and does not require fine-grained tuning:
> > >
> > > - **Complex Tasks**: High-dimensional locomotion tasks (e.g., Humanoid) benefit from a moderately larger $\rho$ (e.g., 10–20). This confirms that for complex tasks, the default entropy constraint is indeed a bottleneck, and increasing $B$ unlocks higher performance.
> > > - **Simple Tasks**: For lower-dimensional tasks (e.g., Pusher), performance is insensitive to $\rho$. Importantly, adjusting $\rho$ does not degrade performance, as the scores remain stable and near-optimal.
> > >
> > > In summary, **$\rho$ acts as a safe and effective hyperparameter: it offers significant potential gains in complex environments without negatively impacting simpler ones**.

---

> > > > ### Comment · Reviewer_Z1ff · 2025-11-26
> > > >
> > > > Thank the authors for the explanation! I get the logic that $\rho$ increases will make performance bound higher. This, however, is against what you provide in the previous response: *This aligns with our empirical observation that performance first improves and then degrades as $\rho$ increases for the Humanoid task*. Can you clarify this?

---

> > > > > ### Author Response · Authors · 2025-11-26
> > > > >
> > > > > We appreciate your careful reading! We would like to clarify that the positive and negative effects dominate in different regimes as $\rho$ increases:
> > > > >
> > > > > - **Improvement Regime**: Initially, the benefit of a higher bound dominates, leading to the performance gains ($11382 \rightarrow 12118 \rightarrow 12542$ ) observed from $\rho=1 \rightarrow 10 \rightarrow 20$.
> > > > >
> > > > > - **Degradation Regime**: The negative impact of reduced exploration becomes critical only when $\rho$ becomes excessively large. At $\rho=30$, the entropy budget becomes too restrictive, overshadowing the benefit of the higher bound and causing the performance drop ($11747$).  While this result is a marginal decline from the peak performance, it remains within a highly competitive range, demonstrating the tuning stability of $\rho$.
> > > > >
> > > > > To conclude, the performance drop at $\rho=30$ after peaking at $\rho=20$ is specific to the excessively large $\rho$ setting, which aligns with our analysis that: *a properly chosen higher $\rho$ can improve the performance, whereas an excessively large $\rho$ (corresponding to an overly negative entropy budget) could reduce exploration and lead to a performance drop.*

---

> > > > > > ### Comment · Reviewer_Z1ff · 2025-11-27
> > > > > >
> > > > > > Thank the authors for their patient explanations. I am still not fully convinced, and I feel that the conclusion *a properly chosen higher $\rho$ can improve performance, whereas an excessively large $\rho$ (corresponding to an overly negative entropy budget) could reduce exploration and lead to a performance drop* actually weakens the theoretical analysis in Section 3.3. Therefore, I recommend that the authors revise the title of Section 3.3 to something like "Empirical Explanation of the Performance Bound" to more accurately reflect the nature of the argument.
> > > > > >
> > > > > > Overall, I am satisfied with the authors' response and raise my score to 6. I will leave it to the AC to make the final decision, taking into account the feedback from all reviewers.

---

> > > > > > > ### Author Response · Authors · 2025-11-27
> > > > > > > **Thanks**
> > > > > > >
> > > > > > > We sincerely thank you for the constructive feedback and for raising the score to 6. We fully accept your suggestion regarding Section 3.3. We agree that the discussion involves empirical interpretation of the bound, and revising the title to "Empirical Explanation of the Performance Bound" (or a similar phrasing) will more accurately reflect the nature of our analysis. **We will incorporate this change in the final version of the paper. We appreciate your support!**

---

> ### Author Response · Authors · 2025-11-21
>
> ## W3&Q3: Explanation of negative entropy budget
> > Why is $\mathcal{H}_{\text{budget}}$ stated to be negative in Sec. 4.1?
>
> The policy entropy is computed as $\mathcal{H}(\pi(\cdot \mid s)) = -\mathbb{E}\_{a\sim\pi}[\log \pi(a\mid s)]$. This term is generally negative when the policy variance is relatively small (the likelihood is larger than 1), which is common for most continuous control tasks. Thus the discounted cumulative policy entropy $\sum\_{t=0}^\infty \gamma^t \mathcal{H}(\pi(\cdot\mid s_t))$ is also generally negative.
>
> Regarding the defined entropy targets: since $\mathcal{H}\_0$ is specified as $-\mathrm{dim}(\mathcal{A}) < 0$ and $1-\gamma > 0$ (given $\gamma \in (0,1)$ in RL), it follows directly that $\mathcal{H}\_{\text{budget}} = \frac{\rho\,\mathcal{H}\_0}{1-\gamma} < 0$.
>
>
> ## Q4: Clarification for environment results
> > Some experimental results (e.g., Reacher-v2) differ from the DSAC-T paper. Could you explain the source of these discrepancies and the reason for evaluating only eight environments?
>
> Thanks for your careful review. In fact, the difference between our reported result $(−3.1 \pm 0.2)$ and the original DSAC-T result $(-3 \pm 0)$ is solely due to decimal rounding. Since the performance on this task is somewhat saturated, retaining one additional decimal place helps better distinguish small performance differences. We hope this clarification resolves your concern.
>
> To address the concern of limited experimental scope, we have extended our evaluation to 4 additional tasks from diverse benchmarks, and detailed results can be seen in the response to **Reviewer 7f3h**.
>
> ## Final thanks
> Thank you again for your time, effort, and professionalism. We hope our responses address your questions and concerns. We are happy to provide additional details if needed, and we look forward to further discussion.

---

### Official Review · Reviewer_z3ry · 2025-10-29

**Soundness:** 3
**Presentation:** 2
**Contribution:** 2
**Rating:** 6
**Confidence:** 4

**Summary:**

The paper proposes Trajectory Entropy-Constrained RL (TECRL), which decouples reward and entropy learning by training two critics: a reward critic $Q_r$ and an entropy critic $Q_e$. The entropy critic estimates cumulative future entropy so the policy can be optimized under a trajectory-level entropy constraint. Building on DSAC-T, the authors instantiate DSAC-E and claim higher stability and returns on 8 MuJoCo tasks. They also present a simple performance-bound argument showing how choosing the entropy budget affects the attainable return, plus ablations for the two components (reward-entropy separation and the trajectory entropy constraint) and sensitivity to the entropy-budget scale $\rho$.

**Strengths:**

- The proposed technique is simple and clearly presented, with writing that is easy to follow.
- TECRL offers significant performance gains compared to the DSAC-T baseline, especially for certain control tasks such as Humanoid, Ant, and Walker2d.

**Weaknesses:**

- The method introduces an additional hyperparameter, $\rho$, which appears to require environment-specific tuning (e.g., $\rho=20$ for Humanoid/Walker2d vs. $\rho=1$ elsewhere), increasing tuning complexity.

- The theoretical investigation is limited. The “performance bound” (Sec. 3.3) fixes $\alpha_\text{soft}^{\*}$ from the MaxEnt optimum and algebraically relates return to $\mathcal{H}^{\*}\_{\text{soft}} - \mathcal{H}\_{\text{budget}}$. However, it doesn’t yield a constructive guarantee for selecting $\rho$ or quantify optimality.

- Results are limited to classic MuJoCo tasks; there is no validation on harder/modern continuous-control suites or high-dimensional, contact-rich domains.

**Minor Error:**
- “we extends maximum entropy framework” → “we extend the maximum-entropy framework.” in Line 17.

**Questions:**

- In MaxEnt RL, the fixed-point iteration yields $\pi(a|s) \propto \exp(\frac{1}{\alpha} Q(s,a))$. Could the authors provide a theoretical justification for convergence under TECRL? Specifically: (i) under what conditions does $Q_e$ converge? (ii) what is the resulting closed-form (or fixed-point) characterization of the optimal $\pi$ in TECRL?
- $H_0$ in Line 260 appears undefined. Is it $\mathcal{H}_0$ introduced in Eq. (6)? Please standardize the notation.
- The choice of $\rho$ seems nontrivial and varies from 1 to 20 across tasks. When should a larger vs. smaller $\rho$ be preferred? Please provide a practical tuning guideline (e.g., ranges, scaling with horizon/entropy targets) and include analyses on additional environments to demonstrate robustness.
- To inform $\rho$ selection, could the authors report the policy’s action variance (or entropy) over training and at evaluation time, and relate these trends to returns and constraint satisfaction?
- Could the authors analyze compute? What’s the end-to-end throughput and wall-clock cost of adding $Q_e$ (forward/backward breakdown)?
- TECRL shows large gains on Humanoid, Ant, and Walker2d but only parity elsewhere. Which task properties (e.g., horizon, contact complexity, multimodality, reward shaping, exploration difficulty) drive the improvements? A brief failure-mode analysis would be helpful.

---

> ### Author Response · Authors · 2025-11-21
>
> We sincerely appreciate your detailed and constructive feedback. Below are our responses to your comments.
>
> ## Q1: Theoretical analysis of TECRL
> > In MaxEnt RL, the fixed-point iteration yields $\pi(a|s) \propto \exp(\frac{1}{\alpha} Q(s,a))$. Could the authors provide a theoretical justification for convergence under TECRL? Specifically: (i) under what conditions does $Q_e$ converge? (ii) what is the resulting closed-form (or fixed-point) characterization of the optimal $\pi$ in TECRL?
>
>
> To clarify the potential misunderstanding, we have re-derived the theoretical foundations of TECRL, with the aim of providing a more intuitive understanding of its underlying objective:
>
> ### 1.1 The convergence of $Q_e$
> We first establish the convergence of $Q_e$ and detailed derivations have been provided in Appendix A.2. The key idea is to construct a Bellman equation for $Q_e$ by analogy with policy evaluation on $Q_r$.
> $(\mathcal{T}\_e Q)(s,a)=-\gamma \log \pi(a \mid s)+\gamma \mathbb{E}_{s',a' \sim p,\pi} \[ Q(s',a') \]$.
>
> We analyze the infinity norm of $\mathcal{T}\_e Q$ operator. For any two functions $Q_1(s, a)$ and $Q_2(s, a)$, we have:
>
> $|| \mathcal{T}\_e Q\_1 - \mathcal{T}\_e Q\_2 ||\_\infty = \gamma * || \mathbb{E}_{s',a'}[ Q\_1(s',a') - Q\_2(s',a') ] ||\_\infty \le \gamma * || Q_1 - Q_2 ||\_\infty$.
>
> It can be seen that $\mathcal{T}\_e$ is a $\gamma$-contraction under the sup-norm, satisfying the Banach fixed-point theorem. Therefore, iteratively applying $\mathcal{T}_e$ to any initial function $Q_0$ converges to a unique fixed point $Q\_e^*$, which satisfies the Bellman equation:
>
> $Q\_e^\star(s,a)=-\gamma \log \pi(a \mid s)+\gamma \mathbb{E}\_{s',a' \sim p,\pi} [ Q\_e^\star(s',a') ].$
>
> ### 1.2 The closed form of optimal policy under TECRL
>
> In maximum entropy RL, the Boltzmann distribution of value function is as follows:
> $p^\star(a|s)=\frac{1}{Z(s)}\exp\left(\frac{Q(s,a)}{\alpha}\right),$
> The core idea of TECRL is to introduce a constraint that the trajectory entropy equals a predefined entropy budget $\mathcal{H}\_\text{budget}$. So the problem can be expressed as:
> $ \max\_{\pi}  \mathbb{E}\_{\pi} \Big[ \sum_{t=0}^\infty \gamma^t [r(s_t,a_t) + \alpha \mathcal{H}(\pi(\cdot|s\_t))] \Big]$
>   s.t.
> $\mathbb{E}\_{\pi} \Big[ \sum\_{t=0}^\infty \gamma^t \mathcal{H}(\pi(\cdot|s\_t)) \Big] = \mathcal{H}\_{\text{budget}}. $
>
> In fact, introducing a trajectory-entropy constraint is essentially equivalent to using a weight function $w(s,a)\propto\exp\left( Q\_e(s,a)\right)$ to refine the original Boltzmann distribution:
>
> $\hat{p}(a\mid s)\propto\underbrace{\exp\left(\frac{1}{\alpha}Q\_r(s,a)\right)}_{\text{reward Boltzmann}}\cdot\underbrace{w(s,a)}\_{\text{TEC-refinement}}=\exp\left(\frac{1}{\alpha}Q\_r(s,a)+Q\_e(s,a)\right)$.
>
> So the target of TECRL can be derived as an approximation between the policy and the refined distribution:
> $\mathrm{KL}(\pi\|\hat{p})=\int\pi(a|s)\log\frac{\pi(a|s)}{\hat{p}(a|s)}da$
>
> $=\int\pi(a|s)\log\pi(a|s)da-\int\pi(a|s)\log \hat{p}(a|s)da$
>
> $=-\mathcal{H}(\pi(\cdot|s))-\mathbb{E}\_{a\sim\pi}\left[\log \hat{p}(a|s)\right]$
>
> $=-\mathcal{H}(\pi(\cdot|s))-\mathbb{E}\_{a\sim\pi}\left[\frac{1}{\alpha}Q(s,a)+Q_e(s,a)-\log Z(s)\right]$
>
> $=-\mathcal{H}(\pi(\cdot|s))-\frac{1}{\alpha}\mathbb{E}\_\pi[Q_r(s,a)+ \alpha Q_e(s,a)]+\log Z(s).$
>
> Finally we get the training policy objective of in our paper:
> $J_{\pi}= Q_r(s,a)+ \alpha  Q_e(s,a) + \alpha \mathcal{H}(\pi(\cdot|s)).$
>
> To conclude, the resulting closed-form characterization of the optimal $\pi$ in TECRL is ${\pi^*}(a\mid s)\propto\exp\left(\frac{1}{\alpha}Q_r(s,a)+Q_e(s,a)\right)$.
>
> ## Q2: Standardize the notation
> > $H_0$ in Line 260 appears undefined. Is it $\mathcal{H}_0$ introduced in Eq. (6)? Please standardize the notation.
>
> Thank you for pointing out the notation inconsistency regarding $H_0$ in original Line 260. You are correct that $H_0$ refers to $\mathcal{H}_0$ introduced in Eq. (6); this was a typographical oversight in our initial draft. To resolve this, we have standardized the notation throughout the manuscript: all references to the single-step entropy target now consistently use $\mathcal{H}_0$, aligning with the definition in Eq. (6). We apologize for the confusion caused by the inconsistency and appreciate your careful review.

---

> ### Author Response · Authors · 2025-11-21
>
> ## W1&Q3&W3: Clarification and experimental illustration of tuning $\rho$
> > The method introduces an additional hyperparameter $\rho$, which appears to require environment-specific tuning (e.g., for Humanoid/Walker2d vs. elsewhere), increasing tuning complexity.
>
> > The choice of $\rho$ seems nontrivial and varies from 1 to 20 across tasks. When should a larger vs. smaller be preferred? Please provide a practical tuning guideline (e.g., ranges, scaling with horizon/entropy targets) and include analyses on additional environments to demonstrate robustness.
>
> > The “performance bound” (Sec. 3.3) fixes $\alpha\_\text{soft}^{\star}$ from the MaxEnt optimum and algebraically relates return to $\mathcal{H}^{*}\_{\text{soft}} - \mathcal{H}\_{\text{budget}}$. However, it doesn’t yield a constructive guarantee for selecting $\rho$ or quantify optimality.
>
> We appreciate your concerns regarding the tuning complexity of the additional hyperparameter $\rho$ and the lack of constructive guidelines.
>
> To address this, we conducted ablation studies on $\rho$ across two representative tasks (Humanoid and Pusher) and summarize practical tuning principles below. The results are shown in [[Figure06_ablation_on_rho]](https://anonymous.4open.science/r/ICLR2026_DSACE_FIG-BBD7/Figure06_ablation_on_rho.pdf) (Figure 13 in the revised paper) and the numerical results are listed in the following Table:
>
> | Algorithm      | ρ = 1         | ρ = 10        | ρ = 20        | ρ = 30        |
> |----------------|----------------|----------------|----------------|----------------|
> | **Humanoid** | 11382 ± 447  | 12118 ± 505 | 12542 ± 280 | 11747 ± 365 |
> | **Pusher**   | -17.7 ± 0.2  |-17.6 ± 0.4  |-17.6 ± 0.3  |-18.0 ± 0.5  |
>
> In practice, the final performance maintains or improves with different $ρ$ values, indicating that $\rho$ does not require fine-grained tuning.
> - High-dimensional and fall-prone locomotion tasks (Humanoid) can benefit from moderately larger $\rho$ (e.g., 10–20). Starting from 1 and gradually increasing $\rho$ is likely to yield a higher return.
> - Relatively simple low-dimensional tasks are not much affected by changes in $\rho$, but it is worth noting that the performance remains close to the optimal level, suggesting that adjusting $\rho$ is unlikely to bring negative impacts.
>
> Overall, $\rho$ remains easy to tune, highly robust across tasks, and does not alter the qualitative behavior or performance advantages of TECRL.
>
> ## Q4: Visualization of policy entropy with varying $\rho$
> > To inform $\rho$ selection, could the authors report the policy’s action variance (or entropy) over training and at evaluation time, and relate these trends to returns and constraint satisfaction?
>
> We have added policy-entropy curves and present it with the return curve together, as shown in [[Figure07_viz_policy_entropy]](https://anonymous.4open.science/r/ICLR2026_DSACE_FIG-BBD7/Figure07_viz_policy_entropy.pdf) (Figure 14 in the revised paper), to more clearly illustrate the role of $\rho$ and the behavior of the entropy budget in TECRL.
>
> The curves verify that the realized policy entropy closely matches the prescribed entropy budget throughout training for a wide range (1~20). $\rho$ = 30 is excessively large and the constraint is not well satisfied. We also observe that increasing $\rho$ from 20 to 30 results in a slight performance degradation as expected.
>
> Overall, performance first improves and then degrades as $\rho$ increases, which aligns with our theoretical analysis: a properly chosen entropy budget can lift the performance bound, whereas an excessively large $\rho$ (corresponding to an overly small entropy budget) reduces exploration and leads to a performance drop.
>
>
> ## Q5: Time-efficiency of TECRL
> > Could the authors analyze compute? What’s the end-to-end throughput and wall-clock cost of adding
>  (forward/backward breakdown)?
>
> To further analyze the time-efficiency of the TECRL, we compare the overall wall-clock time of both DSAC-T (maximum entropy) and DSAC-E (TECRL) in three tasks. The results are as follows:
>
> | Algorithm   | Humanoid (376,17) | Ant (111,8)     | Walker2d (17,6) |
> |-------------|--------------------|------------------|------------------|
> | **DSAC-T**  | 2h09m (129)        | 1h47m (107)      | 1h29m (89)       |
> | **DSAC-E**  | 2h23m (143)        | 2h02m (122)      | 1h44m (104)      |
> | **Percentage** | **110.8%**      | **114.0%**       | **116.9%**       |
>
> All experiments are conducted on a single NVIDIA RTX 3090ti GPU and AMD Ryzen Threadripper 3960X 24-Core Processor. The programming framework is Jax 0.4.28. These results demonstrate that our DSAC-E only introduces minimal computational overhead due to the $Q_e$ update. The overall time overhead is around 10-17%, which is acceptable considering the significant performance improvements achieved by TECRL.

---

> ### Author Response · Authors · 2025-11-21
>
> ## Q6: Task performance analysis
> > TECRL shows large gains on Humanoid, Ant, and Walker2d but only parity elsewhere. Which task properties (e.g., horizon, contact complexity, multimodality, reward shaping, exploration difficulty) drive the improvements? A brief failure-mode analysis would be helpful.
>
> We argue that TECRL enables more accurate critic learning due to reward-entropy separation (RES) and adaptive entropy allocation due to trajectory entropy constraint (TEC). **For high-dimensional and fall-prone locomotion tasks such as Humanoid, Ant, or Walker, TECRL demonstrates clear advantages by avoiding falls and discovering richer behaviors.**
>
> - For Walker and Humanoid, thanks to the adaptive entropy allocation, TECRL exhibits **higher entropy in the middle CoM height range (normal states) and lower entropy at extreme CoM heights (dangerous states)** as shown [[Figure02_entropy_evidence]](https://anonymous.4open.science/r/ICLR2026_DSACE_FIG-BBD7/Figure02_entropy_evidence.pdf) (Figure 8 in the revised paper), consequently better avoiding termination and achiving better performance.
> - For Ant, TECRL learns **coordinated four-leg locomotion and achieves better performance**, whereas baselines such as DSAC-T typically converge to suboptimal three-leg gaits, results are illustrated in [[Figure08_demo_ant]](https://anonymous.4open.science/r/ICLR2026_DSACE_FIG-BBD7/Figure08_demo_ant.pdf) (Figure 15 in the revised paper).
>
> As for **simpler low-dimensional tasks such as Reacher and InvertedDoublePendulum, the gains are much less pronounced because maximum-entropy RL alone is sufficient to reach the performance ceiling**, as shown in [[Figure09_demo_reacher]](https://anonymous.4open.science/r/ICLR2026_DSACE_FIG-BBD7/Figure09_demo_reacher.pdf) (Figure 16 in the revised paper) and [[Figure10_demo_idp]](https://anonymous.4open.science/r/ICLR2026_DSACE_FIG-BBD7/Figure10_demo_idp.pdf) (Figure 17 in the revised paper).
>
> ## W3: Expand experimental scope
> > Results are limited to classic MuJoCo tasks; there is no validation on harder/modern continuous-control suites or high-dimensional, contact-rich domains.
>
> To address the concern of limited experimental scope, we have extended our evaluation to four diverse tasks from modern benchmarks (including DMC’s challenging locomotion tasks, robotic manipulation, and visual-input driving tasks), as detailed in our response to **Reviewer 7f3h**. These new experiments further validate the robustness and broad applicability of TECRL across various benchmarks and control modalities.
>
> ## Minor comments
> > “we extends maximum entropy framework” → “we extend the maximum-entropy framework.” in Line 17.
>
> Thank you for pointing out this grammatical error. We have corrected it in the revised manuscript.
>
> ## Final thanks
> Thank you again for your time, effort, and professionalism. We hope our responses address your questions and concerns. We are happy to provide additional details if needed, and we look forward to further discussion.

---

> > ### Comment · Reviewer_z3ry · 2025-11-24
> >
> > Thank you to the authors for the response and the additional experiments. My questions have been properly addressed. I lean toward acceptance and will keep my current score.

---

> > > ### Author Response · Authors · 2025-11-27
> > > **Thanks**
> > >
> > > We sincerely thank you for the follow-up. We are pleased that the additional experiments and explanations have addressed your concerns. Your constructive feedback has helped us significantly improve the quality and clarity of our paper. We appreciate your support for acceptance!

---

### Official Review · Reviewer_7f3h · 2025-10-31

**Soundness:** 3
**Presentation:** 3
**Contribution:** 1
**Rating:** 4
**Confidence:** 4

**Summary:**

This paper introduces a new framework, Trajectory Entropy-Constrained Reinforcement Learning (TECRL), to address two perceived "bottlenecks" in standard maximum entropy RL algorithms like Soft Actor-Critic (SAC). The identified bottlenecks are: (1) "non-stationary Q-value estimation," which the authors claim is caused by the temperature parameter alpha being updated simultaneously with the Q-function, thus destabilizing the Bellman target; and (2) "short-sighted local entropy tuning," which only constrains the current single-step entropy rather than the cumulative entropy over a whole trajectory.

To solve this, TECRL proposes two main changes, Reward-Entropy Separation (RES) and Trajectory Entropy Constraint (TEC). The authors instantiate this framework in a practical algorithm called DSAC-E, which builds on the state-of-the-art DSAC-T. Experiments on 8 MuJoCo tasks show that DSAC-E outperforms DSAC-T and other strong baselines.

**Strengths:**

- The idea of completely decoupling the reward and entropy value streams into two separate critics ( $Q_r$ and $Q_e$ ) is a novel and clean architectural approach.
- The proposed algorithm, DSAC-E, demonstrates state-of-the-art performance on a suite of standard MuJoCo benchmarks, consistently outperforming its direct predecessor, DSAC-T, as well as SAC and other baselines.
- The ablation study in Table 2 effectively isolates the performance contributions of the two main components (RES and TEC), showing that each one adds value to the final algorithm.

**Weaknesses:**

1. The paper's entire motivation rests on solving two "bottlenecks," but the justification for their existence and severity is weak and not supported by evidence.
- Non-stationary Q-value: The paper asserts that updating alpha makes the Q-target non-stationary. While true, this is a minor effect compared to the policy pi and Q-function Q themselves being updated, which is the primary source of non-stationarity in all bootstrapped RL. The paper provides no empirical evidence (e.g., plots of target value variance) to prove that the changing alpha is a significant source of instability that actually hinders performance in SOTA methods.
- Short-sighted local tuning: The paper claims local entropy tuning is a fundamental flaw, yet it provides no evidence that this mechanism is a primary bottleneck. The very success of SAC and DSAC-T, which use this "flawed" mechanism, suggests it is a highly effective and robust heuristic. The motivation feels more like a post-hoc justification for a new method rather than a response to a well-documented problem.

2. Limited Experimental Scope: The empirical validation is not general enough. All 8 environments are standard continuous control tasks from the MuJoCo/OpenAI Gym benchmark. While DSAC-E shows strong performance here, these tasks are very similar in nature (locomotion/simple manipulation). To make a convincing case for the general superiority of the TECRL framework, it would need to be validated on a more diverse set of domains, such as pixel-based control (e.g., DMControl), tasks with sparse rewards, or more complex robotics manipulation challenges.

**Questions:**

- Can you provide direct evidence (e.g., plots of Bellman error or target value variance over time) to support the claim that Q-value estimation is significantly non-stationary in SAC/DSAC-T (due to alpha) and that your RES method actually produces more stable Q-value estimates?
- The paper claims TEC allows for the "strategic distribution" of entropy. Can you provide an analysis (e.g., visualizing policy entropy in different parts of the state space) to show this is happening, rather than the mechanism simply finding a better average entropy level?
- Why was the experimental evaluation limited to MuJoCo? Have you considered testing the generality of TECRL on other domains, such as pixel-based tasks, where the exploration challenge is different?

---

> ### Author Response · Authors · 2025-11-21
>
> We sincerely appreciate your detailed and constructive feedback. Below are our responses to your comments.
> ## W1&Q1: Evidence for non-stationary Q-value
> > Non-stationary Q-value: The paper asserts that updating alpha makes the Q-target non-stationary. While true, this is a minor effect compared to the policy pi and Q-function Q themselves being updated, which is the primary source of non-stationarity in all bootstrapped RL. The paper provides no empirical evidence (e.g., plots of target value variance) to prove that the changing alpha is a significant source of instability that actually hinders performance in SOTA methods.
>
> > Can you provide direct evidence (e.g., plots of Bellman error or target value variance over time) to support the claim that Q-value estimation is significantly non-stationary in SAC/DSAC-T (due to alpha) and that your RES method actually produces more stable Q-value estimates?
>
> We acknowledge that the bootstrapping update mechanism of Q-values contributes significantly to the non-stationarity in RL. In this context, we highlight that the coupling of reward and entropy is another crucial contributing factor, and our method can effectively address and eliminate this factor.
>
> As shown in [[Figure01_Q_evidence]](https://anonymous.4open.science/r/ICLR2026_DSACE_FIG-BBD7/Figure01_evidence.pdf) (Figure 7 in the revised paper), we provide direct empirical evidence via Q-loss and Q-std curves (comparing DSAC-T and our DSAC-E) on the Humanoid task:
>
> - Q-loss: DSAC-T exhibits a sharp initial peak and substantial late-stage oscillations. In contrast, DSAC-E maintains stable Q-loss throughout training. Its slight late-stage increase mainly aligns with the growing magnitude of Q-values as task performance improves, rather than instability.
> - Q-std: DSAC-T shows a significantly higher initial peak in Q-std. During the mid-to-late training phases, our DSAC-E’s Q-std remains consistently lower than that of DSAC-T, indicating more stable Q-value estimates.
>
> We emphasize that the observed differences in Q-loss and Q-std between DSAC-T and DSAC-E are mainly attributed to the RES’s elimination of alpha-related non-stationarity, while all other components (policy update, NN structure, etc.) are kept consistent. **This controlled comparison provides a direct evidence for non-stationary Q-value and directly links the improved stability to our proposed RES.**
> ## W2&Q2: Evidence for short-sighted local tuning
> > Short-sighted local tuning: The paper claims local entropy tuning is a fundamental flaw, yet it provides no evidence that this mechanism is a primary bottleneck. The very success of SAC and DSAC-T, which use this "flawed" mechanism, suggests it is a highly effective and robust heuristic. The motivation feels more like a post-hoc justification for a new method rather than a response to a well-documented problem.
>
> > The paper claims TEC allows for the "strategic distribution" of entropy. Can you provide an analysis (e.g., visualizing policy entropy in different parts of the state space) to show this is happening, rather than the mechanism simply finding a better average entropy level?
>
> We acknowledge that SAC and DSAC-T have achieved notable success with their local entropy tuning, but our experiments reveal this mechanism can be improved by equipping the entropy adaptation capability that our TEC aims to provide. To substantiate this and validate the "strategic distribution" of entropy, we conducted targeted analyses on the Humanoid and Walker2d tasks.
>
> We evaluated 50 episodes per task for both DSAC-T and our DSAC-E, then visualized policy entropy distribution across the state space. As shown in [[Figure02_entropy_evidence]](https://anonymous.4open.science/r/ICLR2026_DSACE_FIG-BBD7/Figure02_entropy_evidence.pdf)  (Figure 8 in the revised paper), for intuitive presentation, we set:
> - X-axis: Robot’s center-of-mass (CoM) height (a key, interpretable state variable, and extreme values (too low/high) trigger termination as dangerous states).
> - Y-axis: Sum of action standard deviations, which positively correlates with entropy for diagonal Gaussian policies.
>
> Key observations from the visualizations:
> - DSAC-E consistently exhibits **higher entropy in the middle CoM height range (normal states) and lower entropy at extreme CoM heights (dangerous states)**.
> - DSAC-T shows no such state-aware adaptation, its entropy distribution is relatively uniform (compare to ours) across the entire CoM height range, failing to adjust to state-specific risks.
>
> This confirms that TEC does not merely optimize average entropy. Instead, it enables strategic entropy allocation: preserving exploration in normal states while reducing randomness in high-risk regions to avoid termination. This aligns with the core motivation of addressing local tuning’s short-sightedness. Local tuning (as in DSAC-T) treats all states equally, whereas TEC restrict the policy entropy at the trajectory level, allowing adaptive exploration allocation.

---

> ### Author Response · Authors · 2025-11-21
>
> ## W3&Q3: Expand experimental scope
> > Limited Experimental Scope: The empirical validation is not general enough. All 8 environments are standard continuous control tasks from the MuJoCo/OpenAI Gym benchmark. While DSAC-E shows strong performance here, these tasks are very similar in nature (locomotion/simple manipulation). To make a convincing case for the general superiority of the TECRL framework, it would need to be validated on a more diverse set of domains, such as pixel-based control (e.g., DMControl), tasks with sparse rewards, or more complex robotics manipulation challenges.
>
> > Why was the experimental evaluation limited to MuJoCo? Have you considered testing the generality of TECRL on other domains, such as pixel-based tasks, where the exploration challenge is different?
>
> First, the reason why we use 8 MuJoCo continuous control tasks is that they are among the most widely used and classical benchmarks in RL literature, serving as the standard evaluation setting for nearly all state-of-the-art RL methods (e.g., SAC, DSAC-T). Our initial validation on these tasks ensures direct comparability with existing work for demonstrating the core advantages of TECRL.
>
> As you suggested, we have added 4 more tasks from diverse domains, as shown in [[Figure03_additional_env_intro]](https://anonymous.4open.science/r/ICLR2026_DSACE_FIG-BBD7/Figure03_additional_env_intro.pdf) (Figure 9 in the revised paper), to strengthen the experimental evaluation of our method:
> - **Dog-walk and Dog-run**: Two most challenging locomotion tasks in DMC.
> - **Pusher**: A robotic manipulation task.
> - **Carracing**: A visual-input driving task.
>
> For Pusher and Carracing, we compare our DSAC-E with all baselines. The figures are in [[Figure04_results_pusher_carracing]](https://anonymous.4open.science/r/ICLR2026_DSACE_FIG-BBD7/Figure04_results_pusher_carracing.pdf) (Figure 10 in the revised paper). The numerical results are as follows:
> | Algorithm      | Pusher (manipulation)     | Carracing (visual-input)  |
> |----------------|------------------|------------------|
> | **DSAC-E (ours)** | **-17.7 ± 0.2**     | **918 ± 3**          |
> | DSAC-T       | -19.1 ± 0.5     | 903 ± 7          |
> | DSAC-v1      | -25.6 ± 1.3     | 890 ± 12         |
> | SAC          | -19.6 ± 0.3     | -12 ± 8          |
> | TD3          | -21.4 ± 1.2     | -89 ± 5          |
> | DDPG         | -30.5 ± 6.1     | -93 ± 0          |
> | TRPO         | -23.0 ± 2.2     | 363 ± 323        |
> | PPO          | -22.9 ± 1.4     | 776 ± 98         |
>
> For DMC tasks, we only compare with DSAC-T, as it performs best among all baselines, and other baselines are highly unlikely to train a useful policy for such difficult tasks. The figures are in [Figure05_results_dmc](https://anonymous.4open.science/r/ICLR2026_DSACE_FIG-BBD7/Figure05_results_dmc.pdf) (Figure 11 in the revised paper). The numerical results are as follows:
> | Algorithm | dmc_dog_run       | dmc_dog_walk       |
> |-----------|--------------------|---------------------|
> | **DSAC-E (ours)** | **179.6 ± 6.0**    | **286.9 ± 18.3**    |
> | DSAC-T | 138.3 ± 8.3        | 253.3 ± 18.2        |
>
> Across all four environments, our method consistently outperforms the baselines. Even in scenarios where the baseline already approaches its performance ceiling, TECRL demonstrates clear advantages in sample efficiency and stability, achieving comparable or superior returns with fewer interactions. These cross-benchmark results further validate the robustness and broad applicability of TECRL, highlighting its substantial potential for improving the performance across diverse domains and control modalities.
>
> ## Final thanks
> Thank you again for your time, effort, and professionalism. We hope our responses address your questions and concerns. We are happy to provide additional details if needed, and we look forward to further discussion.

---

> > ### Author Response · Authors · 2025-11-27
> > **Kind Reminder**
> >
> > We would like to express our gratitude again for your valuable feedback.
> >
> > To fully address your concerns, we have carefully provided direct evidence backing our two motivations. Moreover, we have significantly expanded our evaluation benchmark. As suggested, we have conducted additional experiments on complex locomotion, robotic manipulation, and visual-input driving, which further demonstrate the effectiveness of our approach.
> >
> > With the discussion window closing soon, we respectfully hope you could check our detailed response. We remain fully available to answer any more questions you may have.

---

### Author Response · Authors · 2025-12-03

Dear Area Chair and Reviewers,

Thank you for your engagement in reviewing our paper and for providing the valuable feedback.
# Contribution
In summary, our main contributions are as follows:
- **Identification of Bottlenecks**: We identify two critical bottlenecks in conventional maximum entropy RL: (1) non-stationary $Q$-value learning and (2) short-sighted entropy tuning. To address these, we propose the TECRL (Trajectory Entropy Constraint RL) framework, including two techniques: Reward-Entropy Separation (RES) and Trajectory Entropy Constraint (TEC).
- **Methodological Innovation**: We introduce RES to decouple reward and entropy signals, eliminating non-stationarity in $Q$-value learning. Furthermore, we extend the maximum-entropy RL framework by introducing a TEC via the dedicated entropy critic to overcome local tuning issue. It enables the agent to dynamically allocate randomness across the trajectory. As a result, the exploration level of can be adaptively adjusted.
- **Theoretical Analysis**:
We provide rigorous theoretical analysis demonstrating that adjusting the trajectory entropy budget could yield a higher performance bound.
- **Practical Algorithm and Empirical Validation**: We develop DSAC-E, a practical instantiation of our framework. Extensive experiments on complex continuous control tasks demonstrate that DSAC-E significantly outperforms state-of-the-art baselines.

# Reviews
Our initial scores are 6 (**`z3ry`**), 4(**`Z1ff`**), 4(**`7f3h`**), 2(**`H9bY`**). After the rebuttal, two reviewers updated feedback positively.

- **Reviewer `z3ry`:** Confirmed all questions were addressed, maintained a 6 rating (supportive of acceptance), and publicly clarified TECRL’s orthogonality to S2AC (rebutting **Reviewer `H9bY`’s** misunderstanding concern).
- **Reviewer `Z1ff`:** Raised the rating from 4 to 6 after resolving theoretical ambiguities (before the data leak accident).
- **Reviewer `7f3h`:** The primary concern regards the empirical evidence for the two bottlenecks we identified. To address this, we have provided direct empirical support via Q-loss and Q-std curves, as well as visualizations of the policy entropy distribution across the state space. We believe these results provide sufficient evidence to attribute the improved stability and effectiveness directly to our method.
- **Reviewer `H9bY`:** We believe that the initial reject rating stemmed from misinterpreting TECRL’s relationship to S2AC. We clarified they address orthogonal problems (S2AC: policy representation; TECRL: Q-value learning/entropy allocation).  Comparative experiments and clarifications in the rebuttal resolved this core misunderstanding. **Reviewer `z3ry`** publicly confirmed this orthogonality and supported our claim.
# Rebuttal Efforts
To address all reviewers’ concerns, we implemented targeted actions:
-  **Empirical Evidence for Non-stationary Q-value (`7f3h`):**
DSAC-E (ours) showed stable Q-loss (no late oscillations) and lower Q-std vs. the baselines, confirming our RES mitigates alpha-induced instability.
- **Empirical Evidence for Short-sighted Entropy Tuning (`7f3h`):**
DSAC-E (ours) demonstrated strategic entropy distribution via state-space visualizations (Humanoid/Walker2d): higher entropy to normal states and lower entropy to high-risk regions (vs. uniform distribution of baselines).
- **Expanded Experiments (`7f3h`, `z3ry`):**
Added 4 diverse tasks (DMC Dog-walk/Dog-run, Pusher manipulation, Carracing visual-input), where DSAC-E (ours) outperformed all baselines, proving cross-domain robustness.
- **Theoretical & Technical Clarifications  (`z3ry`, `Z1ff`):**
Provided formal convergence proof (in Rebuttal and Appendix A.2) via Banach fixed-point theorem, and derived the optimal policy closed-form.
- **Resolved $\rho$ Tuning Concerns  (`z3ry`, `Z1ff`):**
Ablations (Humanoid/Pusher) showed $\rho$ is easy-to-use and effective (no fine-tuning needed).
- **Writing and Refinement (`7f3h`, `Z1ff`, `H9bY`):**
Strengthened the introduction: Distinguished non-stationarity sources (reward-entropy coupling vs. bootstrapping) and explicitly explained limitations of short-sighted local tuning.
- **Addressed S2AC Comparisons and Misunderstanding (`H9bY`):**
Head-to-head experiments showed TECRL outperforms S2AC on locomotion tasks while acknowledging S2AC’s multi-modal policy advantage on multi-goal tasks, highlighting methodological orthogonality (S2AC improves policy representation; TECRL enhances Q-value accuracy/entropy tuning).
# Conclusion
We have thoroughly addressed all initial concerns via empirical validation, theoretical clarification, expanded experiments, and baseline comparisons. The revised submission demonstrates stronger robustness, clearer theoretical grounding, and resolves the key misunderstanding from the initial 2-rated review. We respectfully request the AC’s consideration of updated reviewer feedback and substantive rebuttal improvements. Thanks for your time, effort and professionalism!

---

### Meta-Review · Area_Chair_jP4a · 2026-01-09

**Summary:**

The reviewers initially expressed significant reservations regarding the empirical validation of the paper's core motivations, specifically questioning whether the claimed non-stationary Q-value estimation and short-sighted entropy tuning were actual bottlenecks requiring the proposed solution. There were also substantial concerns about the theoretical soundness of the performance bound derivation, the limited diversity of the experimental benchmarks which were originally restricted to MuJoCo tasks, and the omission of relevant baselines such as S2AC, which raised questions about the method's novelty and comparative effectiveness.

**Reviewer Concerns:**

The rebuttal successfully addressed the majority of the critical concerns, particularly through the inclusion of new experiments on diverse domains like DMC and robotic manipulation which resolved the issue of limited experimental scope, and the provision of Q-loss and policy entropy visualizations that empirically validated the existence of the identified bottlenecks. The theoretical ambiguities regarding the performance bound were clarified as empirical explanations, satisfying Reviewer Z1ff, and the relationship with S2AC was effectively distinguished as orthogonal by the authors and supported by Reviewer z3ry, rendering the novelty concerns largely resolved; consequently, there are no major technical concerns that remain outstanding, as the authors provided the specific evidence and theoretical clarifications requested by the review panel.

**Reviewer Scores:**

Reviewer z3ry maintained a score of 6 and Reviewer Z1ff explicitly raised their score to 6 following the clarifications, indicating a consensus on acceptance among the active participants. Had Reviewer 7f3h participated after the rebuttal, they would likely have raised their score to a 6, given that the authors provided the exact empirical evidence (Q-loss curves and entropy visualizations) and additional experiments they requested to validate the method's motivation. Reviewer H9bY would likely have adjusted their score to 4, as their primary ground for rejection regarding the S2AC baseline was shown to be based on a misunderstanding of methodological orthogonality.

---

### Decision · Program_Chairs · 2026-01-26

Reject